# Twice regularized MDPs and the equivalence between robustness and regularization

**Esther Derman**[*]
Technion

**Matthieu Geist**
Google Research, Brain Team

**Shie Mannor**
Technion, NVIDIA Research

## Abstract

Robust Markov decision processes (MDPs) aim to handle changing or partially known system dynamics. To solve them, one typically resorts to robust optimization methods. However, this significantly increases computational complexity and limits scalability in both learning and planning. On the other hand, regularized MDPs show more stability in policy learning without impairing time complexity. Yet, they generally do not encompass uncertainty in the model dynamics. In this work, we aim to learn robust MDPs using regularization. We first show that regularized MDPs are a particular instance of robust MDPs with uncertain reward. We thus establish that policy iteration on reward-robust MDPs can have the same time complexity as on regularized MDPs. We further extend this relationship to MDPs with uncertain transitions: this leads to a regularization term with an additional dependence on the value function. We finally generalize regularized MDPs to twice regularized MDPs ($R^2$ MDPs), *i.e.*, MDPs with *both* value and policy regularization. The corresponding Bellman operators enable developing policy iteration schemes with convergence and robustness guarantees. It also reduces planning and learning in robust MDPs to regularized MDPs.

## 1 Introduction

MDPs provide a practical framework for solving sequential decision problems under uncertainty [30]. However, the chosen strategy can be very sensitive to sampling errors or inaccurate model estimates. This can lead to complete failure in common situations where the model parameters vary adversarially or are simply unknown [22]. Robust MDPs aim to mitigate such sensitivity by assuming that the transition and/or reward function $(P, r)$ varies arbitrarily inside a given *uncertainty set* $\mathcal{U}$ [17, 27]. In this setting, an optimal solution maximizes a performance measure under the worst-case parameters. It can be thought of as a dynamic zero-sum game with an agent choosing the best action while Nature imposes it the most adversarial model. As such, solving robust MDPs involves max-min problems, which can be computationally challenging and limits scalability.

In recent years, several methods have been developed to alleviate the computational concerns raised by robust reinforcement learning (RL). Apart from [23, 24] that consider specific types of coupled uncertainty sets, all rely on a rectangularity assumption without which the problem can be NP-hard [2, 40]. This assumption is key to deriving tractable solvers of robust MDPs such as robust value iteration [2, 11] or more general robust modified policy iteration (MPI) [18]. Yet, reducing time complexity in robust Bellman updates remains challenging and is still researched today [15, 11].

At the same time, the empirical success of regularization in policy search methods has motivated a wide range of algorithms with diverse motivations such as improved exploration [12, 20] or stability [34, 13]. Geist et al. [10] proposed a unified view from which many existing algorithms can be derived. Their regularized MDP formalism opens the path to error propagation analysis

---

[*]Contact author: `estherderman@campus.technion.ac.il`

35th Conference on Neural Information Processing Systems (NeurIPS 2021).

in approximate MPI [33] and leads to the same bounds as for standard MDPs. Nevertheless, as we further show in Sec. 3, policy regularization accounts for reward uncertainty only: it does not encompass uncertainty in the model dynamics. Despite a vast literature on *how* regularized policy search works and convergence rates analysis [36, 5], little attention has been given to understanding *why* it can generate strategies that are robust to external perturbations [13].

To our knowledge, the only works that relate robustness to regularization in RL are [6, 16, 9]. Derman & Mannor [6] employ a distributionally robust optimization approach to regularize an empirical value function. Unfortunately, computing this empirical value necessitates several policy evaluation procedures, which is quickly unpractical. Husain et al. [16] provide a dual relationship with robust MDPs under uncertain reward. Their duality result applies to general regularization methods and gives a robust interpretation of soft-actor-critic [13]. Although these two works justify the use of regularization for ensuring robustness, they do not enclose any algorithmic novelty. Similarly, Eysenbach & Levine [9] specifically focus on maximum entropy methods and relate them to either reward or transition robustness. We shall further detail on these most related studies in Sec. 7.

The robustness-regularization duality is well established in statistical learning theory [41, 35, 19], as opposed to RL theory. In fact, standard setups such as classification or regression may be considered as single-stage decision-making problems, *i.e.*, one-step MDPs, a particular case of RL setting. Extending this robustness-regularization duality to RL would yield cheaper learning methods with robustness guarantees. As such, we introduce a regularization function $\Omega_{\mathcal{U}}$ that depends on the uncertainty set $\mathcal{U}$ and is defined over both policy and value spaces, thus inducing a *twice regularized* Bellman operator (see Sec. 5). We show that this regularizer yields an equivalence of the form $v_{\pi,\mathcal{U}} = v_{\pi,\Omega_{\mathcal{U}}}$, where $v_{\pi,\mathcal{U}}$ is the robust value function for policy $\pi$ and $v_{\pi,\Omega_{\mathcal{U}}}$ the regularized one. This equivalence is derived through the objective function each value optimizes. More concretely, we formulate the robust value function $v_{\pi,\mathcal{U}}$ as an optimal solution of the robust optimization problem:

$$\max_{v \in \mathbb{R}^{\mathcal{S}}} \langle v, \mu_0 \rangle \text{ s. t. } v \leq \inf_{(P,r) \in \mathcal{U}} T^{\pi}_{(P,r)} v, \tag{RO}$$

where $T^{\pi}_{(P,r)}$ is the evaluation Bellman operator [30]. Then, we show that $v_{\pi,\mathcal{U}}$ is also an optimal solution of the convex (non-robust) optimization problem:

$$\max_{v \in \mathbb{R}^{\mathcal{S}}} \langle v, \mu_0 \rangle \text{ s. t. } v \leq T^{\pi}_{(P_0,r_0)} v - \Omega_{\mathcal{U}}(\pi, v), \tag{CO}$$

where $(P_0, r_0)$ is the *nominal model*. This establishes equivalence between the two optimization problems. Moreover, the inequality constraint of (CO) enables to derive a *twice regularized* ($R^2$) Bellman operator defined according to $\Omega_{\mathcal{U}}$, a policy and value regularizer. For ball-constrained uncertainty sets, $\Omega_{\mathcal{U}}$ has an explicit form and under mild conditions, the corresponding $R^2$ Bellman operators are contracting. The equivalence between the two problems (RO) and (CO) together with the contraction properties of $R^2$ Bellman operators enable to circumvent robust optimization problems at each Bellman update. As such, it alleviates robust planning and learning algorithms by reducing them to regularized ones, which are known to be as complex as classical methods.

To summarize, we make the following contributions: (i) We show that regularized MDPs are a specific instance of robust MDPs with uncertain reward. Besides formalizing a general connection between the two settings, our result enables to explicit the uncertainty sets induced by standard regularizers. (ii) We extend this duality to MDPs with uncertain transition and provide the first regularizer that recovers robust MDPs with $s$-rectangular balls and arbitrary norm. (iii) We introduce twice regularized MDPs ($R^2$ MDPs) that apply both policy and value regularization to retrieve robust MDPs. We establish contraction of the corresponding Bellman operators. This leads us to proposing an $R^2$ MPI algorithm with similar time complexity as vanilla MPI, thus opening new perspectives towards practical and scalable robust RL.

**Notations.** We designate the extended reals by $\overline{\mathbb{R}} := \{-\infty, \infty\}$. Given a finite set $\mathcal{Z}$, the class of real-valued functions (resp. probability distributions) over $\mathcal{Z}$ is denoted by $\mathbb{R}^{\mathcal{Z}}$ (resp. $\Delta_{\mathcal{Z}}$), while the constant function equal to 1 over $\mathcal{Z}$ is denoted by $\mathbb{1}_{\mathcal{Z}}$. Similarly, for any set $\mathcal{X}$, $\Delta_{\mathcal{Z}}^{\mathcal{X}}$ denotes the class of functions defined over $\mathcal{X}$ and valued in $\Delta_{\mathcal{Z}}$. The inner product of two functions $\mathbf{a}, \mathbf{b} \in \mathbb{R}^{\mathcal{Z}}$ is defined as $\langle \mathbf{a}, \mathbf{b} \rangle := \sum_{z \in \mathcal{Z}} \mathbf{a}(z) \mathbf{b}(z)$, which induces the $\ell_2$-norm $\|\mathbf{a}\| := \sqrt{\langle \mathbf{a}, \mathbf{a} \rangle}$. The $\ell_2$-norm coincides with its dual norm, *i.e.*, $\|\mathbf{a}\| = \max_{\|\mathbf{b}\| \leq 1} \langle \mathbf{a}, \mathbf{b} \rangle =: \|\mathbf{a}\|_*$. Let a function $f : \mathbb{R}^{\mathcal{Z}} \to \overline{\mathbb{R}}$. The Legendre-Fenchel transform (or convex conjugate) of $f$ is $f^*(\mathbf{y}) := \max_{\mathbf{a} \in \mathbb{R}^{\mathcal{Z}}} \{\langle \mathbf{a}, \mathbf{y} \rangle - f(\mathbf{a})\}$. Given a set $\mathfrak{Z} \subseteq \mathbb{R}^{\mathcal{Z}}$, the characteristic function $\delta_{\mathfrak{Z}} : \mathbb{R}^{\mathcal{Z}} \to \overline{\mathbb{R}}$ is $\delta_{\mathfrak{Z}}(\mathbf{a}) = 0$ if $\mathbf{a} \in \mathfrak{Z}$; $+\infty$ otherwise. The Legendre-Fenchel transform of $\delta_{\mathfrak{Z}}$ is the support function $\sigma_{\mathfrak{Z}}(\mathbf{y}) = \max_{\mathbf{a} \in \mathfrak{Z}} \langle \mathbf{a}, \mathbf{y} \rangle$ [3, Ex. 1.6.1].

## 2 Preliminaries

This section describes the background material that we use throughout our work. Firstly, we recall useful properties in convex analysis. Secondly, we address classical discounted MDPs and their linear program (LP) formulation. Thirdly, we briefly detail on regularized MDPs and the associated operators and lastly, we focus on the robust MDP setting.

**Convex Analysis.** Let $\Omega : \Delta_{\mathcal{Z}} \to \mathbb{R}$ be a strongly convex function. Throughout this study, the function $\Omega$ plays the role of a policy and/or value regularization function. Its Legendre-Fenchel transform $\Omega^*$ satisfies several smoothness properties, hence its alternative name "smoothed max operator" [25]. Our work makes use of the following result [14, 25].

**Proposition 2.1.** *Given $\Omega : \Delta_{\mathcal{Z}} \to \mathbb{R}$ strongly convex, the following properties hold:*
*(i) $\nabla \Omega^*$ is Lipschitz and satisfies $\nabla \Omega^*(\mathbf{y}) = \arg\max_{\mathbf{a} \in \Delta_{\mathcal{Z}}} \langle \mathbf{a}, \mathbf{y} \rangle - \Omega(\mathbf{a}), \forall \mathbf{y} \in \mathbb{R}^{\mathcal{Z}}.$*
*(ii) For any $c \in \mathbb{R}, \mathbf{y} \in \mathbb{R}^{\mathcal{Z}}, \Omega^*(\mathbf{y} + c\mathbb{1}_{\mathcal{Z}}) = \Omega^*(\mathbf{y}) + c.$*
*(iii) The Legendre-Fenchel transform $\Omega^*$ is non-decreasing.*

**Discounted MDPs and LP formulation.** Consider an infinite horizon MDP $(\mathcal{S}, \mathcal{A}, \mu_0, \gamma, P, r)$ with $\mathcal{S}$ and $\mathcal{A}$ finite state and action spaces respectively, $0 < \mu_0 \in \Delta_{\mathcal{S}}$ an initial state distribution and $\gamma \in (0, 1)$ a discount factor. Denoting $\mathcal{X} := \mathcal{S} \times \mathcal{A}$, $P \in \Delta_{\mathcal{S}}^{\mathcal{X}}$ is a transition kernel mapping each state-action pair to a probability distribution over $\mathcal{S}$ and $r \in \mathbb{R}^{\mathcal{X}}$ is a reward function. A policy $\pi \in \Delta_{\mathcal{A}}^{\mathcal{S}}$ maps any state $s \in \mathcal{S}$ to an action distribution $\pi_s \in \Delta_{\mathcal{A}}$, and we evaluate its performance through the following measure:

$$\rho(\pi) := \mathbb{E}\left[\sum_{t=0}^{\infty} \gamma^t r(s_t, a_t) \;\middle|\; \mu_0, \pi, P\right] = \langle v_{(P,r)}^{\pi}, \mu_0 \rangle. \tag{1}$$

Here, the expectation is conditioned on the process distribution determined by $\mu_0, \pi$ and $P$, and for all $s \in \mathcal{S}$, $v_{(P,r)}^{\pi}(s) = \mathbb{E}[\sum_{t=0}^{\infty} \gamma^t r(s_t, a_t)|s_0 = s, \pi, P]$ is the *value function* at state $s$. Maximizing (1) defines the standard RL objective, which can be solved thanks to the Bellman operators:

$$T_{(P,r)}^{\pi} v := r^{\pi} + \gamma P^{\pi} v \quad \forall v \in \mathbb{R}^{\mathcal{S}}, \pi \in \Delta_{\mathcal{A}}^{\mathcal{S}},$$

$$T_{(P,r)} v := \max_{\pi \in \Delta_{\mathcal{A}}^{\mathcal{S}}} T_{(P,r)}^{\pi} v \quad \forall v \in \mathbb{R}^{\mathcal{S}},$$

$$\mathcal{G}_{(P,r)}(v) := \{\pi \in \Delta_{\mathcal{A}}^{\mathcal{S}} : T_{(P,r)}^{\pi} v = T_{(P,r)} v\} \quad \forall v \in \mathbb{R}^{\mathcal{S}},$$

where $r^{\pi} := [\langle \pi_s, r(s, \cdot) \rangle]_{s \in \mathcal{S}}$ and $P^{\pi} = [P^{\pi}(s'|s)]_{s', s \in \mathcal{S}}$ with $P^{\pi}(s'|s) := \langle \pi_s, P(s'|s, \cdot) \rangle$. Both $T_{(P,r)}^{\pi}$ and $T_{(P,r)}$ are $\gamma$-contractions with respect to (w.r.t.) the supremum norm, so each admits a unique fixed point $v_{(P,r)}^{\pi}$ and $v_{(P,r)}^*$, respectively. The set of greedy policies w.r.t. value $v$ defines $\mathcal{G}_{(P,r)}(v)$, and any policy $\pi \in \mathcal{G}_{(P,r)}(v_{(P,r)}^*)$ is optimal [30]. For all $v \in \mathbb{R}^{\mathcal{S}}$, the associated function $q \in \mathbb{R}^{\mathcal{X}}$ is given by $q(s, a) = r(s, a) + \gamma \langle P(\cdot|s, a), v \rangle \quad \forall (s, a) \in \mathcal{X}$. In particular, the fixed point $v_{(P,r)}^{\pi}$ satisfies $v_{(P,r)}^{\pi} = \langle \pi_s, q_{(P,r)}^{\pi}(s, \cdot) \rangle$ where $q_{(P,r)}^{\pi}$ is its associated $q$-function.

The problem in (1) can also be formulated as an LP [30]. Given a policy $\pi \in \Delta_{\mathcal{A}}^{\mathcal{S}}$, we characterize its performance $\rho(\pi)$ by the following $v$-LP [30, 26]:

$$\min_{v \in \mathbb{R}^{\mathcal{S}}} \langle v, \mu_0 \rangle \text{ subject to (s.t.) } v \geq r^{\pi} + \gamma P^{\pi} v. \tag{$P^{\pi}$}$$

This primal objective provides a policy view on the problem. Alternatively, one may take a state visitation perspective by studying the dual objective instead:

$$\max_{\mu \in \mathbb{R}^{\mathcal{S}}} \langle r^{\pi}, \mu \rangle \text{ s. t. } \mu \geq 0 \text{ and } (\mathbf{Id}_{\mathbb{R}^{\mathcal{S}}} - \gamma P_*^{\pi})\mu = \mu_0, \tag{$D^{\pi}$}$$

where $P_*^{\pi}$ is the *adjoint policy transition operator*[2]: $[P_*^{\pi}\mu](s) := \sum_{\bar{s} \in \mathcal{S}} P^{\pi}(s|\bar{s})\mu(\bar{s}) \quad \forall \mu \in \mathbb{R}^{\mathcal{S}}$, and $\mathbf{Id}_{\mathcal{S}}$ is the identity function in $\mathbb{R}^{\mathcal{S}}$. Let $\mathbf{I}(s'|s, a) := \delta_{s'=s} \quad \forall (s, a, s') \in \mathcal{X} \times \mathcal{S}$ the trivial transition matrix and define its *adjoint transition operator* as $\mathbf{I}_*\mu(s) := \sum_{(\bar{s},\bar{a}) \in \mathcal{X}} \mathbf{I}(s|\bar{s}, \bar{a})\mu(\bar{s}, \bar{a}) \quad \forall s \in$

---

[2]It is the adjoint operator of $P^{\pi}$ in the sense that $\langle P^{\pi} v, v' \rangle = \langle v, P_*^{\pi} v' \rangle \quad \forall v, v' \in \mathbb{R}^{\mathcal{S}}$.

$\mathcal{S}$. The correspondence between occupancy measures and policies lies in the one-to-one mapping $\mu \mapsto \frac{\mu(\cdot, \cdot)}{\mathbf{1}_* \mu(\cdot)} =: \pi_\mu$ and its inverse $\pi \mapsto \mu_\pi$ given by

$$\mu_\pi(s, a) := \sum_{t=0}^{\infty} \gamma^t \mathbb{P} \left( s_t = s, a_t = a \middle| \mu_0, \pi, P \right) \quad \forall (s, a) \in \mathcal{X}.$$

As such, one can interchangeably work with the primal LP ($\mathrm{P}^\pi$) or the dual ($\mathrm{D}^\pi$).

**Regularized MDPs.** A regularized MDP is a tuple $(\mathcal{S}, \mathcal{A}, \mu_0, \gamma, P, r, \Omega)$ with $(\mathcal{S}, \mathcal{A}, \mu_0, \gamma, P, r)$ an infinite horizon MDP as above, and $\Omega := (\Omega_s)_{s \in \mathcal{S}}$ a finite set of functions such that for all $s \in \mathcal{S}, \Omega_s : \Delta_{\mathcal{A}} \to \mathbb{R}$ is strongly convex. Each function $\Omega_s$ plays the role of a policy regularizer $\Omega_s(\pi_s)$. With a slight abuse of notation, we shall denote by $\Omega(\pi) := (\Omega_s(\pi_s))_{s \in \mathcal{S}}$ the family of state-dependent regularizers.[3] The regularized Bellman evaluation operator is given by

$$[T_{(P,r)}^{\pi,\Omega} v](s) := T_{(P,r)}^{\pi} v(s) - \Omega_s(\pi_s) \quad \forall v \in \mathbb{R}^{\mathcal{S}}, s \in \mathcal{S},$$

and the regularized Bellman optimality operator by $T_{(P,r)}^{*,\Omega} v := \max_{\pi \in \Delta_{\mathcal{A}}^{\mathcal{S}}} T_{(P,r)}^{\pi,\Omega} v \quad \forall v \in \mathbb{R}^{\mathcal{S}}$ [10]. The unique fixed point of $T_{(P,r)}^{\pi,\Omega}$ (respectively $T_{(P,r)}^{*,\Omega}$) is denoted by $v_{(P,r)}^{\pi,\Omega}$ (resp. $v_{(P,r)}^{*,\Omega}$) and defines the *regularized value function* (resp. *regularized optimal value function*). Although the regularized MDP formalism stems from the aforementioned Bellman operators in [10], it turns out that regularized MDPs are MDPs with modified reward. Indeed, for any policy $\pi \in \Delta_{\mathcal{A}}^{\mathcal{S}}$, the regularized value function is $v_{(P,r)}^{\pi,\Omega} = (\mathbf{I}_{\mathcal{S}} - \gamma P^\pi)^{-1}(r^\pi - \Omega(\pi))$, which corresponds to a non-regularized value with expected reward $\tilde{r}^\pi := r^\pi - \Omega(\pi)$. Note that the modified reward $\tilde{r}^\pi(s)$ is no longer linear in $\pi_s$ because of the strong convexity of $\Omega_s$. Also, this modification does not apply to the reward function $r$ but only to its expectation $r^\pi$, as we cannot regularize the original reward without making it policy-independent.

**Robust MDPs.** In general, the MDP model is not explicitly known but rather estimated from sampled trajectories. As this may result in over-sensitive outcome [22], robust MDPs reduce such performance variation. Formally, a robust MDP $(\mathcal{S}, \mathcal{A}, \mu_0, \gamma, \mathcal{U})$ is an MDP with uncertain model belonging to $\mathcal{U} := \mathcal{P} \times \mathcal{R}$, *i.e.*, uncertain transition $P \in \mathcal{P} \subseteq \Delta_{\mathcal{S}}^{\mathcal{X}}$ and reward $r \in \mathcal{R} \subseteq \mathbb{R}^{\mathcal{X}}$ [17, 40]. The uncertainty set $\mathcal{U}$ typically controls the confidence level of a model estimate, which in turn determines the agent's level of robustness. It is given to the agent, who seeks to maximize performance under the worst-case model $(P, r) \in \mathcal{U}$. Although untractable in general, this problem can be solved in polynomial time for *rectangular* uncertainty sets, *i.e.*, when $\mathcal{U} = \times_{s \in \mathcal{S}} \mathcal{U}_s = \times_{s \in \mathcal{S}} (\mathcal{P}_s \times \mathcal{R}_s)$ [40, 23]. For any policy $\pi \in \Delta_{\mathcal{A}}^{\mathcal{S}}$ and state $s \in \mathcal{S}$, the *robust value function* at $s$ is $v^{\pi,\mathcal{U}}(s) := \min_{(P,r) \in \mathcal{U}} v_{(P,r)}^{\pi}(s)$ and the *robust optimal value function* $v^{*,\mathcal{U}}(s) := \max_{\pi \in \Delta_{\mathcal{A}}^{\mathcal{S}}} v^{\pi,\mathcal{U}}(s)$. Each of them is the unique fixed point of the respective robust Bellman operators:

$$[T^{\pi,\mathcal{U}} v](s) := \min_{(P,r) \in \mathcal{U}} T_{(P,r)}^{\pi} v(s) \quad \forall v \in \mathbb{R}^{\mathcal{S}}, s \in \mathcal{S}, \pi \in \Delta_{\mathcal{S}}^{\mathcal{A}},$$

$$[T^{*,\mathcal{U}} v](s) := \max_{\pi \in \Delta_{\mathcal{A}}^{\mathcal{S}}} [T^{\pi,\mathcal{U}} v](s) \quad \forall v \in \mathbb{R}^{\mathcal{S}}, s \in \mathcal{S},$$

which are $\gamma$-contractions. For all $v \in \mathbb{R}^{\mathcal{S}}$, the associated robust $q$-function is given by $q(s, a) = \min_{(P,r) \in \mathcal{U}} \{r(s, a) + \gamma \langle P(\cdot|s, a), v \rangle\} \quad \forall (s, a) \in \mathcal{X}$, so that $v^{\pi,\mathcal{U}} = \langle \pi_s, q^{\pi,\mathcal{U}}(s, \cdot) \rangle$ where $q^{\pi,\mathcal{U}}$ is the robust $q$-function associated to $v^{\pi,\mathcal{U}}$.

## 3 Reward-robust MDPs

This section focuses on reward-robust MDPs, *i.e.*, robust MDPs with uncertain reward but known transition model. We first show that regularized MDPs represent a particular instance of reward-robust MDPs, as both solve the same optimization problem. This equivalence provides a theoretical motivation for the heuristic success of policy regularization. Then, we explicit the uncertainty set

---

[3]In the formalism of Geist et al. [10], $\Omega_s$ is initially constant over $\mathcal{S}$. However, later in the paper [10, Sec. 5], it changes according to policy iterates. Here, we alternatively define a family $\Omega$ of state-dependent regularizers, which accounts for state-dependent uncertainty sets (see Sec. 5 below).

underlying some standard regularization functions, thus suggesting an interpretable explanation of their empirical robustness.

We first show the following proposition 3.1, which applies to general robust MDPs and random policies. It slightly extends [17], as Lemma 3.2 there focuses on uncertain-transition MDPs and deterministic policies. For completeness, we provide a proof of Prop. 3.1 in Appx. A.1.

**Proposition 3.1.** *For any policy* $\pi \in \Delta_{\mathcal{A}}^{\mathcal{S}}$*, the robust value function* $v^{\pi,\mathcal{U}}$ *is the optimal solution of the robust optimization problem:*

$$\max_{v \in \mathbb{R}^{\mathcal{S}}} \langle v, \mu_0 \rangle \; s. \; t. \; v \leq T_{(P,r)}^{\pi} v \; \text{for all} \; (P,r) \in \mathcal{U}. \tag{$P_{\mathcal{U}}$}$$

In the robust optimization problem $(P_{\mathcal{U}})$, the inequality constraint must hold over the whole uncertainty set $\mathcal{U}$. As such, a function $v \in \mathbb{R}^{\mathcal{S}}$ is said to be *robust feasible* for $(P_{\mathcal{U}})$ if $v \leq T_{(P,r)}^{\pi} v$ for all $(P,r) \in \mathcal{U}$ or equivalently, if $\max_{(P,r) \in \mathcal{U}} \{v(s) - T_{(P,r)}^{\pi} v(s)\} \leq 0$ for all $s \in \mathcal{S}$. Therefore, checking robust feasibility requires to solve a maximization problem. For properly structured uncertainty sets, a closed form solution can be derived, as we shall see in the sequel. As standard in the robust RL literature [32, 15, 1], the remaining of this work focuses on uncertainty sets centered around a known *nominal model*. Formally, given $P_0$ (resp. $r_0$) a nominal transition kernel (resp. reward function), we consider uncertainty sets of the form $(P_0 + \mathcal{P}) \times (r_0 + \mathcal{R})$. The size of $\mathcal{P} \times \mathcal{R}$ quantifies our level of uncertainty or alternatively, the desired degree of robustness.

**Reward-robust and regularized MDPs: an equivalence.** We now focus on reward-robust MDPs, *i.e.*, robust MDPs with $\mathcal{U} = \{P_0\} \times (r_0 + \mathcal{R})$. Thm. 3.1 establishes that reward-robust MDPs are in fact regularized MDPs whose regularizer is given by a support function. Its proof can be found in Appx. A.2. This result brings two take-home messages: (i) policy regularization is equivalent to reward uncertainty; (ii) policy iteration on reward-robust MDPs has the same convergence rate as regularized MDPs, which in turn is the same as standard MDPs [10].

**Theorem 3.1** (Reward-robust MDP). *Assume that* $\mathcal{U} = \{P_0\} \times (r_0 + \mathcal{R})$*. Then, for any policy* $\pi \in \Delta_{\mathcal{A}}^{\mathcal{S}}$*, the robust value function* $v^{\pi,\mathcal{U}}$ *is the optimal solution of the convex optimization problem:*

$$\max_{v \in \mathbb{R}^{\mathcal{S}}} \langle v, \mu_0 \rangle \; s. \; t. \; v(s) \leq T_{(P_0,r_0)}^{\pi} v(s) - \sigma_{\mathcal{R}_s}(-\pi_s) \; \text{for all} \; s \in \mathcal{S}.$$

Thm. 3.1 clearly highlights a convex regularizer $\Omega_s(\pi_s) := \sigma_{\mathcal{R}_s}(-\pi_s) \quad \forall s \in \mathcal{S}$. We thus recover a regularized MDP by setting $[T^{\pi,\Omega} v](s) = T_{(P_0,r_0)}^{\pi} v(s) - \sigma_{\mathcal{R}_s}(-\pi_s) \quad \forall s \in \mathcal{S}$. In particular, when $\mathcal{R}_s$ is a ball of radius $\alpha_s^r$, the support function (or regularizer) can be written in closed form as $\Omega_s(\pi_s) := \alpha_s^r \|\pi_s\|$, which is strongly convex. We formalize this below (see proof in Appx. A.3).

**Corollary 3.1.** *Let* $\pi \in \Delta_{\mathcal{A}}^{\mathcal{S}}$ *and* $\mathcal{U} = \{P_0\} \times (r_0 + \mathcal{R})$*. Further assume that for all* $s \in \mathcal{S}$*, the reward uncertainty set at* $s$ *is* $\mathcal{R}_s := \{r_s \in \mathbb{R}^{\mathcal{A}} : \|r_s\| \leq \alpha_s^r\}$*. Then, the robust value function* $v^{\pi,\mathcal{U}}$ *is the optimal solution of the convex optimization problem:*

$$\max_{v \in \mathbb{R}^{\mathcal{S}}} \langle v, \mu_0 \rangle \; s. \; t. \; v(s) \leq T_{(P_0,r_0)}^{\pi} v(s) - \alpha_s^r \|\pi_s\| \; \text{for all} \; s \in \mathcal{S}.$$

While regularization induces reward-robustness, Thm. 3.1 and Cor. 3.1 suggest that, on the other hand, specific reward-robust MDPs recover well-known policy regularization methods. In the following section, we explicit the reward-uncertainty sets underlying some of these regularizers.

**Related Algorithms.** Consider a reward uncertainty set of the form $\mathcal{R} := \times_{(s,a) \in \mathcal{X}} \mathcal{R}_{s,a}$. This defines an $(s,a)$-rectangular $\mathcal{R}$ (a particular type of $s$-rectangular $\mathcal{R}$) whose rectangles $\mathcal{R}_{s,a}$ are independently defined for each state-action pair. For the regularizers below, we derive appropriate $\mathcal{R}_{s,a}$-s that recover the same regularized value function. Detailed proofs are in Appx. A.4. There, we also include a summary table that reviews the properties of some RL regularizers, as well as our $R^2$ function which we shall introduce later in Sec. 5. Note that the reward uncertainty sets here depend on the policy. This is due to the fact that standard regularizers are defined over the policy space and not at each state-action pair. It similarly explains why the reward modification induced by regularization does not apply to the original reward function, as already mentioned in Sec. 2.

*Negative Shannon entropy.* Let $\mathcal{R}_{s,a}^{\text{NS}}(\pi) := [\ln\left(1/\pi_s(a)\right), +\infty) \quad \forall (s,a) \in \mathcal{X}$. The associated support function enables to write:

$$\sigma_{\mathcal{R}_s^{\text{NS}}(\pi)}(-\pi_s) = \max_{r(s,\cdot):r(s,a')\in\mathcal{R}_{s,a'}^{\text{NS}}(\pi),a'\in\mathcal{A}} \sum_{a\in\mathcal{A}} -r(s,a)\pi_s(a) = \sum_{a\in\mathcal{A}} \pi_s(a)\ln(\pi_s(a)),$$

where the last equality comes from maximizing $-r(s,a)$ over $[\ln\left(1/\pi_s(a)\right), +\infty)$ for each $a \in \mathcal{A}$. We thus recover the negative Shannon entropy $\Omega(\pi_s) = \sum_{a\in\mathcal{A}} \pi_s(a)\ln(\pi_s(a))$ [13].

*Kullback-Leibler divergence.* Given an action distribution $0 < d \in \Delta_{\mathcal{A}}$, let $\mathcal{R}_{s,a}^{\text{KL}}(\pi) := \ln\left(d(a)\right) + \mathcal{R}_{s,a}^{\text{NS}}(\pi) \quad \forall (s,a) \in \mathcal{X}$. It amounts to translating the interval $\mathcal{R}_{s,a}^{\text{NS}}$ by the given constant. Similarly writing the support function yields $\Omega(\pi_s) = \sum_{a\in\mathcal{A}} \pi_s(a)\ln\left(\pi_s(a)/d(a)\right)$, which is exactly the KL divergence [34].

*Negative Tsallis entropy.* Letting $\mathcal{R}_{s,a}^{\text{T}}(\pi) := [(1-\pi_s(a))/2, +\infty) \quad \forall (s,a) \in \mathcal{X}$, we recover the negative Tsallis entropy $\Omega(\pi_s) = \frac{1}{2}(\|\pi_s\|^2 - 1)$ [20].

**Policy-gradient for reward-robust MDPs.** The equivalence between reward-robust and regularized MDPs leads us to wonder whether we can employ policy-gradient [37] on reward-robust MDPs using regularization. The following result establishes that a policy-gradient theorem can indeed be established for reward-robust MDPs (see proof in Appx. A.5).

**Proposition 3.2.** *Assume that $\mathcal{U} = \{P_0\} \times (r_0 + \mathcal{R})$ with $\mathcal{R}_s = \{r_s \in \mathbb{R}^{\mathcal{A}} : \|r_s\| \leq \alpha_s^r\}$. Then, the gradient of the reward-robust objective $J_{\mathcal{U}}(\pi) := \langle v^{\pi,\mathcal{U}}, \mu_0 \rangle$ is given by*

$$\nabla J_{\mathcal{U}}(\pi) = \mathbb{E}_{(s,a)\sim\mu_\pi}\left[\nabla \ln \pi_s(a)\left(q^{\pi,\mathcal{U}}(s,a) - \alpha_s^r \frac{\pi_s(a)}{\|\pi_s\|}\right)\right],$$

*where $\mu_\pi$ is the occupancy measure under the nominal model $P_0$ and policy $\pi$.*

Although Prop. 3.2 is an application of [10, Appx. D.3] for a specific regularized MDP, its reward-robust formulation is novel and suggests another simplification of robust methods. Indeed, previous works that exploit policy-gradient on robust MDPs involve the occupancy measure of the worst-case model [21], whereas our result sticks to the nominal. In practice, Prop. 3.2 enables to learn a robust policy by sampling transitions from the nominal model instead of all uncertain models. This has a twofold advantage: (i) it avoids an additional computation of the minimum as done in [29, 21, 7], where the authors sample next-state transitions and rewards based on all parameters from the uncertainty set, then update a policy based on the worst outcome; (ii) it releases from restricting to finite uncertainty sets. In fact, our regularizer accounts for robustness regardless of the sampling procedure, whereas the parallel simulations of [29, 21, 7] require the uncertainty set to be finite. Technical difficulties are yet to be addressed for generalizing our result to transition-uncertain MDPs, because of the interdependence between the regularizer and the value function (see Secs. 4-5). We detail more on this issue in Appx. A.5.

# 4 General robust MDPs

Now that we have established policy regularization as a reward-robust problem, we would like to study the opposite question: can any robust MDP with both uncertain reward and transition be solved using regularization instead of robust optimization? If so, is the regularization function easy to determine? In this section, we answer positively to both questions for properly defined robust MDPs. This greatly facilitates robust RL, as it avoids the increased complexity of robust planning algorithms while still reaching robust performance.

The following theorem establishes that similarly to reward-robust MDPs, robust MDPs can be formulated through regularization (see proof in Appx. B.1). Although the regularizer is also a support function in that case, it depends on both the policy and the value objective, which may further explain the difficulty of dealing with robust MDPs.

**Theorem 4.1** (General robust MDP). *Assume that $\mathcal{U} = (P_0 + \mathcal{P}) \times (r_0 + \mathcal{R})$. Then, for any policy $\pi \in \Delta_{\mathcal{A}}^{\mathcal{S}}$, the robust value function $v^{\pi,\mathcal{U}}$ is the optimal solution of the convex optimization problem:*

$$\max_{v\in\mathbb{R}^{\mathcal{S}}} \langle v, \mu_0 \rangle \ s.\ t.\ v(s) \leq T_{(P_0,r_0)}^{\pi}v(s) - \sigma_{\mathcal{R}_s}(-\pi_s) - \sigma_{\mathcal{P}_s}(-\gamma v \cdot \pi_s) \ \text{for all } s \in \mathcal{S}, \qquad (2)$$

*where $[v \cdot \pi_s](s', a) := v(s')\pi_s(a) \quad \forall (s', a) \in \mathcal{X}$.*

The upper-bound in the inequality constraint (2) is of the same spirit as the regularized Bellman operator: the first term is a standard, non-regularized Bellman operator on the nominal model $(P_0, r_0)$ to which we subtract a policy and value-dependent function playing the role of regularization. This function reminds that of [6, Thm. 3.1] also coming from conjugacy. This is the only similarity between both regularizers: in [6], the Legendre-Fenchel transform is applied on a different type of function and results in a regularization term that has no closed form but can only be bounded from above. Moreover, the setup considered there is different since it studies distributionally robust MDPs. As such, it involves general convex optimization, whereas we focus on the robust formulation of an LP.

The support function further simplifies when the uncertainty set is a ball, as shown below. Yet, the dependence of the regularizer on the value function prevents us from readily applying the tool-set of regularized MDPs. We shall study the properties of this new regularization function in Sec. 5.

**Corollary 4.1.** *Assume that* $\mathcal{U} = (P_0 + \mathcal{P}) \times (r_0 + \mathcal{R})$ *with* $\mathcal{P}_s := \{P_s \in \mathbb{R}^{\mathcal{X}} : \|P_s\| \leq \alpha_s^P\}$ *and* $\mathcal{R}_s := \{r_s \in \mathbb{R}^{\mathcal{A}} : \|r_s\| \leq \alpha_s^r\}$ *for all* $s \in \mathcal{S}$. *Then, the robust value function* $v^{\pi, \mathcal{U}}$ *is the optimal solution of the convex optimization problem:*

$$\max_{v \in \mathbb{R}^{\mathcal{S}}} \langle v, \mu_0 \rangle \ s.\ t.\ v(s) \leq T_{(P_0, r_0)}^{\pi} v(s) - \alpha_s^r \|\pi_s\| - \alpha_s^P \gamma \|v\| \|\pi_s\| \ for\ all\ s \in \mathcal{S}. \tag{3}$$

One can actually take two different norms for the reward and the transition uncertainty sets. Similarly, Cor. 4.1 can be rewritten with an arbitrary norm, which would reveal a dual norm $\|\cdot\|_*$ instead of $\|\cdot\|$ in Eq. (3) (see proof in Appx. B.2). Here, we restrict our statement to the $\ell_2$-norm for notation convenience only, the dual norm of $\ell_2$ being $\ell_2$ itself. Thus, our regularization function recovers a robust value function independently of the chosen norm, which extends previous results from [15, 11]. Indeed, Ho et al. [15] lighten complexity of robust planning for the $\ell_1$-norm only, while Grand-Clément & Kroer [11] focus on KL and $\ell_2$ ball-constrained uncertainty sets. Both works rely on the specific structure induced by the divergence they consider to derive more efficient robust Bellman updates. Differently, our method circumvents these updates using a generic, problem-independent regularization function while still encompassing $s$-rectangular uncertainty sets as in [15, 11].

# 5  R² MDPs

In Sec. 4, we showed that for general robust MDPs, the optimization constraint involves a regularization term that depends on the value function itself. This adds a difficulty to the reward-robust case where the regularization only depends on the policy. Yet, we provided an explicit regularizer for general robust MDPs that are ball-constrained. In this section, we introduce R² MDPs, an extension of regularized MDPs that combines policy and value regularization. The core idea is to further regularize the Bellman operators with a value-dependent term that recovers the support functions we derived from the robust optimization problems of Secs. 3-4.

**Definition 5.1** (R² Bellman operators). *For all* $v \in \mathbb{R}^{\mathcal{S}}$, *define* $\Omega_{v, \mathrm{R}^2} : \Delta_{\mathcal{A}} \to \mathbb{R}$ *as* $\Omega_{v, \mathrm{R}^2}(\pi_s) := \|\pi_s\| (\alpha_s^r + \alpha_s^P \gamma \|v\|)$. *The R² Bellman evaluation and optimality operators are defined as*

$$[T^{\pi, \mathrm{R}^2} v](s) := T_{(P_0, r_0)}^{\pi} v(s) - \Omega_{v, \mathrm{R}^2}(\pi_s) \quad \forall s \in \mathcal{S},$$

$$[T^{*, \mathrm{R}^2} v](s) := \max_{\pi \in \Delta_{\mathcal{A}}^{\mathcal{S}}} [T^{\pi, \mathrm{R}^2} v](s) = \Omega_{v, \mathrm{R}^2}^*(q_s) \quad \forall s \in \mathcal{S}.$$

*For any function* $v \in \mathbb{R}^{\mathcal{S}}$, *the associated unique greedy policy is defined as*

$$\pi_s = \arg \max_{\pi_s \in \Delta_{\mathcal{A}}} T^{\pi, \mathrm{R}^2} v(s) = \nabla \Omega_{v, \mathrm{R}^2}^*(q_s), \quad \forall s \in \mathcal{S},$$

*that is, in vector form,* $\pi = \nabla \Omega_{v, \mathrm{R}^2}^*(q) =: \mathcal{G}_{\Omega_{\mathrm{R}^2}}(v) \iff T^{\pi, \mathrm{R}^2} v = T^{*, \mathrm{R}^2} v$.

The R² Bellman evaluation operator is not linear because of the functional norm appearing in the regularization function. Yet, under the following assumption, it is contracting and we can apply Banach's fixed point theorem to define the R² value function.

**Assumption 5.1** (Bounded radius). *For all* $s \in \mathcal{S}$, *there exists* $\epsilon_s > 0$ *such that*

$$\alpha_s^P \leq \min \left( \frac{1 - \gamma - \epsilon_s}{\gamma \sqrt{|\mathcal{S}|}}; \min_{\substack{\mathbf{u}_{\mathcal{A}} \in \mathbb{R}_+^{\mathcal{A}}, \|\mathbf{u}_{\mathcal{A}}\| = 1 \\ \mathbf{v}_{\mathcal{S}} \in \mathbb{R}_+^{\mathcal{S}}, \|\mathbf{v}_{\mathcal{S}}\| = 1}} \mathbf{u}_{\mathcal{A}}^{\top} P_0(\cdot | s, \cdot) \mathbf{v}_{\mathcal{S}} \right).$$

Asm. 5.1 requires to upper bound the ball radius of transition uncertainty sets. The first term in the minimum is needed for establishing contraction of $R^2$ Bellman operators (item (iii) in Prop. 5.1), while the second one is used for ensuring monotonicity (item (i) in Prop. 5.1). We remark that the former depends on the original discount factor $\gamma$: radius $\alpha_s^P$ must be smaller as $\gamma$ tends to 1 but can arbitrarily grow as $\gamma$ decreases to 0, without altering contraction. Indeed, larger $\gamma$ implies longer time horizon and higher stochasticity, which explains why we need tighter level of uncertainty then. Otherwise, value and policy regularization seem unable to handle the mixed effects of parameter and stochastic uncertainties. The additional dependence on the state-space size comes from the $\ell_2$-norm chosen for the ball constraints. In fact, for any $\ell_p$-norm of dual $\ell_q$, $|\mathcal{S}|^{\frac{1}{q}}$ replaces $\sqrt{|\mathcal{S}|}$ in the denominator, so it becomes independent of $|\mathcal{S}|$ as $(p,q)$ tends to $(1,\infty)$ (see Appx. C.1). Although we recognize a generalized Rayleigh quotient-type problem in the second minimum [28], its interpretation in our context remains unclear. Asm. 5.1 enables the $R^2$ Bellman operators to admit a unique fixed point, among other nice properties. We formalize this below (see proof in Appx. C.1).

**Proposition 5.1.** *Suppose that Asm. 5.1 holds. Then, we have the following properties:*

*(i) Monotonicity: For all $v_1, v_2 \in \mathbb{R}^{\mathcal{S}}$ such that $v_1 \leq v_2$, we have $T^{\pi,R^2} v_1 \leq T^{\pi,R^2} v_2$ and $T^{*,R^2} v_1 \leq T^{*,R^2} v_2$.*

*(ii) Sub-distributivity: For all $v_1 \in \mathbb{R}^{\mathcal{S}}, c \in \mathbb{R}$, we have $T^{\pi,R^2}(v_1 + c\mathbb{1}_{\mathcal{S}}) \leq T^{\pi,R^2} v_1 + \gamma c\mathbb{1}_{\mathcal{S}}$ and $T^{*,R^2}(v_1 + c\mathbb{1}_{\mathcal{S}}) \leq T^{*,R^2} v_1 + \gamma c\mathbb{1}_{\mathcal{S}}, \forall c \in \mathbb{R}$.*

*(iii) Contraction: Let $\epsilon_* := \min_{s \in \mathcal{S}} \epsilon_s > 0$. Then, for all $v_1, v_2 \in \mathbb{R}^{\mathcal{S}}$, $\|T^{\pi,R^2} v_1 - T^{\pi,R^2} v_2\|_\infty \leq (1 - \epsilon_*)\|v_1 - v_2\|_\infty$ and $\|T^{*,R^2} v_1 - T^{*,R^2} v_2\|_\infty \leq (1 - \epsilon_*)\|v_1 - v_2\|_\infty$.*

We should note that the contracting coefficient in Prop. 5.1 is different from the original discount factor $\gamma$, since here we have $1 - \epsilon^*$. Yet, as Asm. 5.1 suggests it, an intrinsic dependence between $\gamma$ and $\epsilon^*$ makes the $R^2$ Bellman updates similar to the standard ones: when $\gamma$ tends to 0, the value of $\epsilon^*$ required for Asm. 5.1 to hold increases, which makes the contracting coefficient $1 - \epsilon^*$ tend to 0 as well, *i.e.*, the two contracting coefficients behave similarly. The contraction of both $R^2$ Bellman operators finally leads us to introduce the $R^2$ value functions.

**Definition 5.2** ($R^2$ value functions). *(i) The $R^2$ value function $v^{\pi,R^2}$ is defined as the unique fixed point of the $R^2$ Bellman evaluation operator: $v^{\pi,R^2} = T^{\pi,R^2} v^{\pi,R^2}$. The associated q-function is $q^{\pi,R^2}(s,a) = r_0(s,a) + \gamma\langle P_0(\cdot|s,a), v^{\pi,R^2}\rangle$. (ii) The $R^2$ optimal value function $v^{*,R^2}$ is defined as the unique fixed point of the $R^2$ Bellman optimal operator: $v^{*,R^2} = T^{*,R^2} v^{*,R^2}$. The associated q-function is $q^{*,R^2}(s,a) = r_0(s,a) + \gamma\langle P_0(\cdot|s,a), v^{*,R^2}\rangle$.*

The monotonicity of the $R^2$ Bellman operators plays a key role in reaching an optimal $R^2$ policy, as we show in the following. A proof can be found in Appx. C.2.

**Theorem 5.1** ($R^2$ optimal policy). *The greedy policy $\pi^{*,R^2} = \mathcal{G}_{\Omega_{R^2}}(v^{*,R^2})$ is the unique optimal $R^2$ policy, i.e., for all $\pi \in \Delta_{\mathcal{A}}^{\mathcal{S}}, v^{\pi^*,R^2} = v^{*,R^2} \geq v^{\pi,R^2}$.*

**Remark 5.1.** *An optimal $R^2$ policy may be stochastic. This is due to the fact that our $R^2$ MDP framework builds upon the general $s$-rectangularity assumption. Robust MDPs with $s$-rectangular uncertainty sets similarly yield an optimal robust policy that is stochastic [40, Table 1]. Nonetheless, the $R^2$ MDP formulation recovers a deterministic optimal policy in the more specific $(s,a)$-rectangular case, which is in accordance with the robust MDP setting (see proof in Appx. C.3).* [4]

All of the results above ensure convergence of MPI in $R^2$ MDPs. We call that method $R^2$ MPI and provide its pseudo-code in Alg. 1. The convergence proof follows the same lines as in [30]. Moreover, the contracting property of the $R^2$ Bellman operator ensures the same convergence rate as in standard and robust MDPs, *i.e.*, a geometric convergence rate. On the other hand, $R^2$ MPI reduces the computational complexity of robust MPI by avoiding to solve a max-min problem at each iteration, as this can take polynomial time for general convex programs. Advantageously, the only optimization involved in $R^2$ MPI lies in the greedy step: it amounts to projecting onto the simplex,

---

**Algorithm 1:** $R^2$ MPI

**Result:** $\pi_{k+1}, v_{k+1}$
Initialize $v_k \in \mathbb{R}^{\mathcal{S}}$;
**while** *not converged* **do**
$\quad \pi_{k+1} \leftarrow \mathcal{G}_{\Omega_{R^2}}(v_k);$
$\quad v_{k+1} \leftarrow (T^{\pi_{k+1},R^2})^m v_k;$
**end**

---

[4]The stochasticity of an optimal entropy-regularized policy as in the examples of Sec. 3 is not contradicting. Indeed, even though the corresponding uncertainty set is $(s,a)$-rectangular there, it is policy-dependent.

which can efficiently be performed in linear time [8]. Still, such projection is not even necessary in the $(s, a)$-rectangular case: as mentioned in Rmk. 5.1, it then suffices to choose a greedy action in order to eventually achieve an optimal $R^2$ value function.

## 6  Numerical Experiments

We aim to compare the computing time of $R^2$ MPI with that of MPI [30] and robust MPI [18]. The code is available at `https://github.com/EstherDerman/r2mdp`. To do so, we run experiments on an Intel(R) Core(TM) i7-1068NG7 CPU @ 2.30GHz machine, which we test on a $5 \times 5$ grid-world domain. In that environment, the agent starts from a random position and seeks to reach a goal state in order to maximize reward. Thus, the reward function is zero in all states but two: one provides a reward of 1 while the other gives 10. An episode ends when either one of those two states is attained.

The partial evaluation of each policy iterate is a building block of MPI. As a sanity check, we evaluate the uniform policy through both $R^2$ and robust policy evaluation (PE) sub-processes, to ensure that the two value outputs coincide. For simplicity, we focus on an $(s, a)$-rectangular uncertainty set and take the same ball radius $\alpha$ (resp. $\beta$) at each state-action pair for the reward function (resp. transition function). Parameter values and other implementation details are deferred to Appx. D. We obtain the same value for $R^2$ PE and robust PE, which numerically confirms Thm. 4.1. On the other hand, both are strictly smaller than their non-robust, non-regularized counterpart, but as expected, they converge to the standard value function when all ball radii tend to 0 (see Appx. D). More importantly, $R^2$ PE converges in 0.02 seconds, whereas robust PE takes 54.8 seconds to converge, *i.e.*, 2740 times longer. This complexity gap comes from the minimization problems being solved at each iteration of robust PE, something that $R^2$ PE avoids thanks to regularization. $R^2$ PE still takes 2.5 times longer than its standard, non-regularized counterpart, because of the additional computation of regularization terms. Table 1 shows the time spent by each algorithm until convergence.

Table 1: Computing time (in sec.) of planning algorithms using vanilla, $R^2$ and robust approaches. Each cell displays the mean $\pm$ standard deviation obtained from 5 running seeds.

|  | **Vanilla** | $\mathbf{R^2}$ | **Robust** |
|---|---|---|---|
| **PE** | $0.008 \pm 0.$ | $0.02 \pm 0.$ | $54.8 \pm 1.2$ |
| **MPI** ($m = 1$) | $0.01 \pm 0.$ | $0.03 \pm 0.$ | $118.6 \pm 1.3$ |
| **MPI** ($m = 4$) | $0.01 \pm 0.$ | $0.03 \pm 0.$ | $98.1 \pm 4.1$ |

We then study the overall MPI process for each approach. We know that in vanilla MPI, the greedy step is achieved by simply searching over deterministic policies [30]. Since we focus our experiments on an $(s, a)$-rectangular uncertainty set, the same applies to robust MPI [40] and to $R^2$ MPI, as already mentioned in Rmk. 5.1. We can see in Table 1 that the increased complexity of robust MPI is even more prominent than its PE thread, as robust MPI takes 3953 (resp. 3270) times longer than $R^2$ MPI when $m = 1$ (resp. $m = 4$). Robust MPI with $m = 4$ is a bit more advantageous than $m = 1$, as it needs less iterations (31 versus 67), *i.e.*, less optimization solvers to converge. Interestingly, for both $m \in \{1, 4\}$, progressing from PE to MPI did not cost much more computing time to either the vanilla or the $R^2$ version: both take less than one second to run.

## 7  Related Work

Connections between regularization and robustness have been established in standard statistical learning settings such as support vector machines [41], logistic regression [35] or maximum likelihood estimation [19]. As stated in Sec. 1, these represent particular RL problems as they concern a single-stage decision-making process. In that regard, the generalization of robustness-regularization duality to sequential decision-making has seldom been studied in the RL literature.

Two works that view policy regularization from a robustness perspective are [16] and [9]. In [16], regularization is applied on the dual objective instead of the primal. This has two shortcomings: (i) It prevents from deriving regularized Bellman operators and dynamic programming methods; (ii) The feasible set is that of occupancy measures, so the connection with standard policy regularization remains unclear. Furthermore, their work focuses on reward robustness. Differently, Eysenbach & Levine [9] address both reward and transition uncertainty by showing that policies with maximum

entropy regularization solve a particular type of robust MDP. Yet, their analysis separately treats the uncertainty on $P$ and $r$, which questions the robustness of the resulting policy when the whole model $(P, r)$ is adversarial. Moreover, the dual relation they establish between entropy regularization and transition-robust MDPs is weak and applies to specific uncertainty sets. Both of these works treat robustness as a side-effect of regularization more than an objective on its own, whereas we aim to do the opposite, namely, use regularization to solve robust RL problems.

Derman & Mannor [6] similarly use regularization as a tool for achieving robust policies. Through distributionally robust MDPs, they show upper and lower bounds between transition-robustness and regularization. There again, duality is weak and reward uncertainty is not addressed. Moreover, since the exact regularization term has no explicit form, it is usable through its upper bound only. Finally, regularization is applied on the mean of several value functions $v^\pi_{(\hat{P}_i, r)}$, where each $\hat{P}_i$ is a transition model estimated from an episode run. Computing this quantity requires as many policy evaluations as the number of model estimates available, which results in a linear complexity blowup at least.

Previous studies analyze robust planning algorithms to provide convergence guarantees. The works [2, 27] propose robust value iteration, while [17, 40] introduce robust policy iteration. Kaufman & Schaefer [18] generalize both schemes by proposing a robust MPI and determine the conditions under which it converges. The polynomial time within which all these works guarantee a robust solution is often insufficient, as the complexity of a Bellman update grows cubically in the number of states [15].

In order to reduce the time complexity of robust planning algorithms, Ho et al. [15] propose two algorithms that compute robust Bellman updates in $\mathcal{O}(|\mathcal{S}||\mathcal{A}| \log(|\mathcal{S}||\mathcal{A}|))$ operations for $\ell_1$-constrained uncertainty sets. Advantageously, our regularization approach reduces each such update to its standard, non-robust complexity of $\mathcal{O}(|\mathcal{S}||\mathcal{A}|)$. Moreover, although Ho et al. [15] address both $(s, a)$ and $s$-rectangular uncertainty sets, they focus on transition uncertainty, whereas we tackle both reward and transition uncertainties in the general $s$-rectangular case. Finally, their contribution relies on LP formulations that necessitate restricting to the $\ell_1$-norm while our method applies to any norm. This may come from the fact that our main Theorems (Thms. 3.1 and 4.1) use Fenchel-Rockafellar duality [31, 4], a generalization of LP duality (see Appx. A.2 and B.1). More recently, Grand-Clément & Kroer [11] propose a first-order method to accelerate robust value iteration under $s$-rectangular uncertainty sets that are either ellipsoidal or KL-constrained. To our knowledge, our study is the first one reducing the time complexity of robust planning when the uncertainty set is $s$-rectangular and constrained with an arbitrary norm.

## 8 Conclusion and future work

In this work, we established a strong duality between robust MDPs and twice regularized MDPs. This revealed that the regularized MDPs of [10] are in fact robust MDPs with uncertain reward, which enabled us to derive a policy-gradient theorem for reward-robust MDPs. When extending this robustness-regularization duality to general robust MDPs, we found that the regularizer depends on the value function besides the policy. We thus introduced $R^2$ MDPs, a generalization of regularized MDPs with both policy and value regularization. The related $R^2$ Bellman operators lead us to propose a converging $R^2$ MPI algorithm that achieves the optimal robust value function within similar computing time as standard MPI.

This study settles the theoretical foundations for scalable robust RL. We should note that our results naturally extend to continuous but compact action spaces in the same manner as standard MDPs do [30]. Extension to infinite state-space would be more involved because of the state-dependent regularizer in $R^2$ MDPs. In fact, it would be interesting to study the $R^2$ MDP setting under function approximation, as such approximation would have a direct effect on the regularizer. Similarly, one could analyze approximate dynamic programming for $R^2$ MDPs in light of its robust analog [38, 1]. Although our theory focused on planning, the $R^2$ Bellman operators and their contracting properties open the path to learning algorithms that can (i) use existing RL algorithms and robustify them by simply changing the regularizer; (ii) scale to deep learning settings. Apart from its practical effect, we believe our work opens the path to more theoretical contributions in robust RL. For example, extending $R^2$ MPI to the approximate case [33] would be an interesting problem to solve because of the $R^2$ evaluation operator being non-linear. So would be a sample complexity analysis for $R^2$ MDPs with a comparison to robust MDPs [42]. Another line of research is to extend policy-gradient to $R^2$ MDPs, as this would avoid parallel learning of adversarial models [7, 39] and be very useful for continuous control.

## Acknowledgements

We would like to thank Raphael Derman for his useful suggestions that improved the clarity of the text. Thanks also to Stav Belogolovsky for reviewing a previous version of this paper. Funding in direct support of this work: ISF grant.

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
