## Appendix: Twice regularized MDPs and the equivalence between robustness and regularization

This appendix provides proofs for all of the results stated in the paper. We first recall the following theorem used in the sequel and referred to as Fenchel-Rochafellar duality [4, Thm 3.3.5].

**Theorem** (Fenchel-Rockafellar duality). *Let $X, Y$ two Euclidean spaces, $f : X \to \overline{\mathbb{R}}$ and $g : Y \to \overline{\mathbb{R}}$ two proper, convex functions, and $A : X \to Y$ a linear mapping such that $0 \in \mathrm{core}(\mathrm{dom}(g) - A(\mathrm{dom}(f)))$.[5] Then, it holds that*

$$\min_{x \in X} f(x) + g(Ax) = \max_{y \in Y} -f^*(-A^*y) - g^*(y). \tag{4}$$

## A  Reward-Robust MDPs

### A.1  Proof of Proposition 3.1

**Proposition.** *For any policy $\pi \in \Delta_{\mathcal{A}}^{\mathcal{S}}$, the robust value function $v^{\pi,\mathcal{U}}$ is the optimal solution of the robust optimization problem:*

$$\max_{v \in \mathbb{R}^{\mathcal{S}}} \langle v, \mu_0 \rangle \ s.\ t.\ v \leq T_{(P,r)}^\pi v \ for\ all\ (P,r) \in \mathcal{U}. \tag{$P_{\mathcal{U}}$}$$

*Proof.* Let $v^*$ an optimal point of ($P_{\mathcal{U}}$). By definition of the robust value function, $v^{\pi,\mathcal{U}} = T^{\pi,\mathcal{U}} v^{\pi,\mathcal{U}} = \min_{(P,r) \in \mathcal{U}} T_{(P,r)}^\pi v^{\pi,\mathcal{U}}$. In particular, $v^{\pi,\mathcal{U}} \leq T_{(P,r)}^\pi v^{\pi,\mathcal{U}}$ for all $(P,r) \in \mathcal{U}$, so the robust value is feasible and by optimality of $v^*$, we get $\langle v^*, \mu_0 \rangle \geq \langle v^{\pi,\mathcal{U}}, \mu_0 \rangle$. Now, we aim to show that any feasible $v \in \mathbb{R}^{\mathcal{S}}$ satisfies $v \leq v^{\pi,\mathcal{U}}$. Let an arbitrary $\epsilon > 0$. Then, there exists $(P_\epsilon, r_\epsilon) \in \mathcal{U}$ such that

$$T^{\pi,\mathcal{U}} v^{\pi,\mathcal{U}} + \epsilon > T_{(P_\epsilon, r_\epsilon)}^\pi v^{\pi,\mathcal{U}}. \tag{5}$$

This yields:

$$
\begin{aligned}
v - v^{\pi,\mathcal{U}} &= v - T^{\pi,\mathcal{U}} v^{\pi,\mathcal{U}} && [v^{\pi,\mathcal{U}} = T^{\pi,\mathcal{U}} v^{\pi,\mathcal{U}}] \\
&< v + \epsilon - T_{(P_\epsilon, r_\epsilon)}^\pi v^{\pi,\mathcal{U}} && [\text{By Eq. (5)}] \\
&\leq T^{\pi,\mathcal{U}} v + \epsilon - T_{(P_\epsilon, r_\epsilon)}^\pi v^{\pi,\mathcal{U}} && [v \text{ is feasible for } (P_{\mathcal{U}})] \\
&\leq T^{\pi,\mathcal{U}} v + \epsilon - T^{\pi,\mathcal{U}} v^{\pi,\mathcal{U}} && [T^{\pi,\mathcal{U}} u = \min_{(P,r) \in \mathcal{U}} T_{(P,r)}^\pi u \leq T_{(P_\epsilon, r_\epsilon)}^\pi u \quad \forall u \in \mathbb{R}^{\mathcal{S}}] \\
&= T^{\pi,\mathcal{U}} (v - v^{\pi,\mathcal{U}}) + \epsilon.
\end{aligned}
$$

Thus, $v - v^{\pi,\mathcal{U}} \leq T^{\pi,\mathcal{U}}(v - v^{\pi,\mathcal{U}}) + \epsilon$, which we iteratively apply as follows:

$$
\begin{aligned}
v - v^{\pi,\mathcal{U}} &\leq T^{\pi,\mathcal{U}}(v - v^{\pi,\mathcal{U}}) + \epsilon \\
&\leq T^{\pi,\mathcal{U}}(T^{\pi,\mathcal{U}}(v - v^{\pi,\mathcal{U}}) + \epsilon) + \epsilon && [\text{By [27]}, u \leq w \implies T^{\pi,\mathcal{U}} u \leq T^{\pi,\mathcal{U}} w] \\
&= (T^{\pi,\mathcal{U}})^2 (v - v^{\pi,\mathcal{U}}) + \gamma \epsilon + \epsilon \\
&\leq (T^{\pi,\mathcal{U}})^2 (T^{\pi,\mathcal{U}}(v - v^{\pi,\mathcal{U}}) + \epsilon) + \gamma \epsilon + \epsilon \\
&\quad \vdots \\
&\leq (T^{\pi,\mathcal{U}})^{n+1}(v - v^{\pi,\mathcal{U}}) + \sum_{k=0}^{n} \gamma^k \epsilon \\
&= (T^{\pi,\mathcal{U}})^{n+1} v - v^{\pi,\mathcal{U}} + \sum_{k=0}^{n} \gamma^k \epsilon. && [v^{\pi,\mathcal{U}} = T^{\pi,\mathcal{U}} v^{\pi,\mathcal{U}}]
\end{aligned}
$$

Setting $n \to \infty$ yields $v - v^{\pi,\mathcal{U}} \leq \frac{\epsilon}{1-\gamma}$. Since both $\epsilon > 0$ and $v$ were taken arbitrarily, $v^* \leq v^{\pi,\mathcal{U}}$, while we have already shown that $\langle v^*, \mu_0 \rangle \geq \langle v^{\pi,\mathcal{U}}, \mu_0 \rangle$. By positivity of the probability distribution $\mu_0$, it results that $\langle v^*, \mu_0 \rangle = \langle v^{\pi,\mathcal{U}}, \mu_0 \rangle$, and since $\mu_0 > 0$, we obtain $v^{\pi,\mathcal{U}} = v^*$. $\square$

---

[5]Given $C \subseteq \mathbb{R}^{\mathcal{S}}$, we say that $x \in \mathrm{core}(C)$ if for all $d \in \mathbb{R}^{\mathcal{S}}$ there exists a small enough $t \in \mathbb{R}$ such that $x + td \in C$ [4].

## A.2 Proof of Theorem 3.1

**Theorem** (Reward-robust MDP). *Assume that $\mathcal{U} = \{P_0\} \times (r_0 + \mathcal{R})$. Then, for any policy $\pi \in \Delta_{\mathcal{A}}^{\mathcal{S}}$, the robust value function $v^{\pi,\mathcal{U}}$ is the optimal solution of the convex optimization problem:*

$$\max_{v \in \mathbb{R}^{\mathcal{S}}} \langle v, \mu_0 \rangle \text{ s. t. } v(s) \leq T_{(P_0,r_0)}^{\pi} v(s) - \sigma_{\mathcal{R}_s}(-\pi_s) \text{ for all } s \in \mathcal{S} .$$

*Proof.* For all $s \in \mathcal{S}$, define: $F(s) := \max_{(P,r) \in \mathcal{U}} \{v(s) - r^{\pi}(s) - \gamma P^{\pi} v(s)\}$. It corresponds to the robust counterpart of $(\mathrm{P}_{\mathcal{U}})$ at $s \in \mathcal{S}$. Thus, the robust value function $v^{\pi,\mathcal{U}}$ is the optimal solution of:

$$\max_{v \in \mathbb{R}^{\mathcal{S}}} \langle v, \mu_0 \rangle \text{ s. t. } F(s) \leq 0 \text{ for all } s \in \mathcal{S} . \tag{6}$$

Based on the structure of the uncertainty set $\mathcal{U} = \{P_0\} \times (r_0 + \mathcal{R})$, we compute the robust counterpart:

$$
\begin{aligned}
F(s) &= \max_{r' \in r_0 + \mathcal{R}} \{v(s) - r'^{\pi}(s) - \gamma P_0^{\pi} v(s)\} \\
&= \max_{r':r'=r_0+r, r\in\mathcal{R}} \{v(s) - r'^{\pi}(s) - \gamma P_0^{\pi} v(s)\} \\
&= \max_{r \in \mathcal{R}} \{v(s) - (r_0^{\pi}(s) + r^{\pi}(s)) - \gamma P_0^{\pi} v(s)\} && [(r_0 + r)^{\pi} = r_0^{\pi} + r^{\pi} \quad \forall \pi \in \Delta_{\mathcal{A}}^{\mathcal{S}}] \\
&= \max_{r \in \mathcal{R}} \{v(s) - r^{\pi}(s) - r_0^{\pi}(s) - \gamma P_0^{\pi} v(s)\} \\
&= \max_{r \in \mathcal{R}} \left\{v(s) - r^{\pi}(s) - T_{(P_0,r_0)}^{\pi} v(s)\right\} && [T_{(P_0,r_0)}^{\pi} v(s) = r_0^{\pi}(s) + \gamma P_0^{\pi} v(s)] \\
&= \max_{r \in \mathcal{R}} \{-r^{\pi}(s)\} + v(s) - T_{(P_0,r_0)}^{\pi} v(s) \\
&= \max_{r \in \mathbb{R}^{\mathcal{X}}} \{-r^{\pi}(s) - \delta_{\mathcal{R}}(r')\} + v(s) - T_{(P_0,r_0)}^{\pi} v(s) \\
&= - \min_{r \in \mathbb{R}^{\mathcal{X}}} \{r^{\pi}(s) + \delta_{\mathcal{R}}(r)\} + v(s) - T_{(P_0,r_0)}^{\pi} v(s) \\
&= - \min_{r \in \mathbb{R}^{\mathcal{X}}} \{\langle r_s, \pi_s \rangle + \delta_{\mathcal{R}}(r)\} + v(s) - T_{(P_0,r_0)}^{\pi} v(s). && [r^{\pi}(s) = \langle r_s, \pi_s \rangle]
\end{aligned}
$$

By the rectangularity assumption, $\mathcal{R} = \times_{s \in \mathcal{S}} \mathcal{R}_s$ and for all $r := (r_s)_{s \in \mathcal{S}} \in \mathbb{R}^{\mathcal{X}}$, we have $\delta_{\mathcal{R}}(r) = \sum_{s' \in \mathcal{S}} \delta_{\mathcal{R}_{s'}}(r_{s'})$. As such,

$$
\begin{aligned}
F(s) &= - \min_{r \in \mathbb{R}^{\mathcal{X}}} \{\langle r_s, \pi_s \rangle + \sum_{s' \in \mathcal{S}} \delta_{\mathcal{R}_{s'}}(r_{s'})\} + v(s) - T_{(P_0,r_0)}^{\pi} v(s) \\
&= - \min_{r \in \mathbb{R}^{\mathcal{X}}} \{\langle r_s, \pi_s \rangle + \delta_{\mathcal{R}_s}(r_s)\} + v(s) - T_{(P_0,r_0)}^{\pi} v(s),
\end{aligned}
$$

where the last equality holds since the objective function is minimal if and only if $r_s \in \mathcal{R}_s$.

We now aim to apply Fenchel-Rockafellar duality to the minimization problem. Let the function $f : \mathbb{R}^{\mathcal{A}} \to \mathbb{R}$ defined as $r_s \mapsto \langle r_s, \pi_s \rangle$, and consider the support function $\delta_{\mathcal{R}_s} : \mathbb{R}^{\mathcal{A}} \to \overline{\mathbb{R}}$ together with the identity mapping $\mathbf{Id}_{\mathcal{A}} : \mathbb{R}^{\mathcal{A}} \to \mathbb{R}^{\mathcal{A}}$. Clearly, $\mathrm{dom}(f) = \mathbb{R}^{\mathcal{A}}$, $\mathrm{dom}(\delta_{\mathcal{R}_s}) = \mathcal{R}_s$, and $\mathrm{dom}(\delta_{\mathcal{R}_s}) - \mathbf{Id}_{\mathcal{A}}(\mathrm{dom}(f)) = \mathcal{R}_s - \mathbb{R}^{\mathcal{A}} = \mathbb{R}^{\mathcal{A}}$. Therefore, $\mathrm{core}(\mathrm{dom}(\delta_{\mathcal{R}_s}) - A(\mathrm{dom}(f))) = \mathrm{core}(\mathbb{R}^{\mathcal{A}}) = \mathbb{R}^{\mathcal{A}}$ and $0 \in \mathbb{R}^{\mathcal{A}}$. We can thus apply Fenchel-Rockafellar duality: noting that $\mathbf{Id}_{\mathcal{A}} = (\mathbf{Id}_{\mathcal{A}})^*$ and $(\delta_{\mathcal{R}_s})^*(y) = \sigma_{\mathcal{R}_s}(y)$, we get

$$\min_{r_s \in \mathbb{R}^{\mathcal{A}}} \{f(r_s) + \delta_{\mathcal{R}_s}(r_s)\} = - \min_{y \in \mathbb{R}^{\mathcal{A}}} \{f^*(-y) + (\delta_{\mathcal{R}_s})^*(y)\} = - \min_{y \in \mathbb{R}^{\mathcal{A}}} \{f^*(-y) + \sigma_{\mathcal{R}_s}(y)\}.$$

It remains to compute

$$f^*(-y) = \max_{r_s \in \mathbb{R}^{\mathcal{A}}} -\langle r_s, y \rangle - \langle r_s, \pi_s \rangle = \max_{r_s \in \mathbb{R}^{\mathcal{A}}} \langle r_s, -y - \pi_s \rangle = \begin{cases} 0 \text{ if } -y - \pi_s = 0 \\ +\infty \text{ otherwise} \end{cases},$$

and obtain

$$F(s) = \min_{y \in \mathbb{R}^{\mathcal{A}}} \{f^*(-y) + \sigma_{\mathcal{R}_s}(y)\} + v(s) - T_{(P_0,r_0)}^{\pi} v(s) = \sigma_{\mathcal{R}_s}(-\pi_s) + v(s) - T_{(P_0,r_0)}^{\pi} v(s).$$

We can thus rewrite the optimization problem (6) as:

$$\max_{v \in \mathbb{R}^{\mathcal{S}}} \langle v, \mu_0 \rangle \text{ s. t. } \sigma_{\mathcal{R}_s}(-\pi_s) + v(s) - T_{(P_0,r_0)}^{\pi} v(s) \leq 0 \text{ for all } s \in \mathcal{S},$$

which concludes the proof. $\qquad\square$

## A.3 Proof of Corollary 3.1

**Corollary.** *Let $\pi \in \Delta_{\mathcal{A}}^{\mathcal{S}}$ and $\mathcal{U} = \{P_0\} \times (r_0 + \mathcal{R})$. Further assume that for all $s \in \mathcal{S}$, the reward uncertainty set at $s$ is $\mathcal{R}_s := \{r_s \in \mathbb{R}^{\mathcal{A}} : \|r_s\| \leq \alpha_s^r\}$. Then, the robust value function $v^{\pi,\mathcal{U}}$ is the optimal solution of the convex optimization problem:*

$$\max_{v \in \mathbb{R}^{\mathcal{S}}} \langle v, \mu_0 \rangle \ s.\ t.\ v(s) \leq T_{(P_0,r_0)}^{\pi} v(s) - \alpha_s^r \|\pi_s\| \ \textit{for all } s \in \mathcal{S}\,.$$

*Proof.* We evaluate the support function:

$$\sigma_{\mathcal{R}_s}(-\pi_s) = \max_{r_s \in \mathbb{R}^{\mathcal{A}}:\|r_s\|\leq\alpha_s^r} \langle r_s, -\pi_s \rangle \overset{(1)}{=} \alpha_s^r\|-\pi_s\| = \alpha_s^r\|\pi_s\|,$$

where equality $(1)$ holds by definition of the dual norm. Applying Thm. 3.1, the robust value function $v^{\pi,\mathcal{U}}$ is the optimal solution of: $\max_{v \in \mathbb{R}^{\mathcal{S}}} \langle v, \mu_0 \rangle$ s. t. $\alpha_s^r\|\pi_s\| + v(s) - T_{(P_0,r_0)}^{\pi} v(s) \leq 0$ for all $s \in \mathcal{S}$, which concludes the proof.

*Ball-constraint with arbitrary norm.* In the case where reward ball-constraints are defined according to an arbitrary norm $\|\cdot\|_a$ with dual norm $\|\cdot\|_{a*}$, the support function becomes:

$$\sigma_{\mathcal{R}_s}(-\pi_s) = \max_{r_s \in \mathbb{R}^{\mathcal{A}}:\|r_s\|_a \leq \alpha_s^r} \langle r_s, -\pi_s \rangle = \alpha_s^r\|-\pi_s\|_{a*} = \alpha_s^r\|\pi_s\|_{a*}.$$

$\square$

## A.4 Related Algorithms: Uncertainty sets from regularizers

Table 2: Summary table of existing policy regularizers and generalization to our $R^2$ function.

| | **Negative Shannon** | **KL divergence** | **Negative Tsallis** | **$R^2$ function** |
|---|---|---|---|---|
| **Regularizer $\Omega$** | $\sum_{a \in \mathcal{A}} \pi_s(a) \ln(\pi_s(a))$ | $\sum_{a \in \mathcal{A}} \pi_s(a) \ln\left(\frac{\pi_s(a)}{d(a)}\right)$ | $\frac{1}{2}(\|\pi_s\|^2 - 1)$ | $\|\pi_s\|(\alpha_s^r + \alpha_s^P \gamma\|v\|)$ |
| **Conjugate $\Omega^*$** | $\ln\left(\sum_{a \in \mathcal{A}} e^{q_s(a)}\right)$ | $\ln\left(\sum_{a \in \mathcal{A}} d(a)e^{q_s(a)}\right)$ | $\frac{1}{2} + \frac{1}{2}\sum_{a \in \mathfrak{A}}(q_s(a)^2 - \tau(q_s)^2)$ | Not in closed-form |
| **Gradient $\nabla\Omega^*$** | $\pi_s(a) = \frac{e^{q_s(a)}}{\sum_{b \in \mathcal{A}} e^{q_s(b)}}$ | $\pi_s(a) = \frac{e^{q_s(a)}}{\sum_{b \in \mathcal{A}} e^{q_s(b)}}$ | $\pi_s(a) = (q_s(a) - \tau(q_s))_+$ | Not in closed-form |
| **Reward Uncertainty** | $(s,a)$-rectangular | $(s,a)$-rectangular | $(s,a)$-rectangular | $s$-rectangular |
| | $\mathcal{R}_{s,a}^{\mathrm{NS}}(\pi) = \left[\ln\left(\frac{1}{\pi_s(a)}\right), +\infty\right)$ | $\ln(d(a)) + \mathcal{R}_{s,a}^{\mathrm{NS}}(\pi)$ | $\left[\frac{1 - \pi_s(a)}{2}, +\infty\right)$ | $\mathbf{B}_{\|\cdot\|}(r_{0s}, \alpha_s^r)$ |
| **Transition Uncertainty** | $(s,a)$-rectangular | $(s,a)$-rectangular | $(s,a)$-rectangular | $s$-rectangular |
| | $\{P_0(\cdot|s,a)\}$ | $\{P_0(\cdot|s,a)\}$ | $\{P_0(\cdot|s,a)\}$ | $\mathbf{B}_{\|\cdot\|}(P_{0s}, \alpha_s^P)$ |

*Negative Shannon entropy.* Each $(s,a)$-reward uncertainty set is $\mathcal{R}_{s,a}^{\mathrm{NS}}(\pi) := [\ln(1/\pi_s(a)), +\infty)$. We compute the associated support function:

$$
\begin{aligned}
\sigma_{\mathcal{R}_s^{\mathrm{NS}}(\pi)}(-\pi_s) &= \max_{r_s \in \mathcal{R}_s^{\mathrm{NS}}(\pi)} \langle r_s, -\pi_s \rangle \\
&= \max_{r(s,a'):r(s,a') \in \mathcal{R}_{s,a'}^{\mathrm{NS}}(\pi), a' \in \mathcal{A}} \sum_{a \in \mathcal{A}} -r(s,a)\pi_s(a) \\
&= \max_{r(s,a'):r(s,a') \geq \ln(1/\pi_s(a)), a' \in \mathcal{A}} -\sum_{a \in \mathcal{A}} \pi_s(a) r(s,a) \\
&= \sum_{a \in \mathcal{A}} \pi_s(a) \ln(\pi_s(a)),
\end{aligned}
\tag{7}
$$

where the last equality results from the fact that $\pi_s \geq 0$, and $-r(s,a)\pi_s(a)$ is maximal when $r(s,a)$ is minimal. We thus obtain the negative Shannon entropy.

*KL divergence.* Similarly, given $d \in \Delta_{\mathcal{A}}$, let $\mathcal{R}_{s,a}^{\mathrm{KL}}(\pi) := \ln(d(a)) + \mathcal{R}_{s,a}^{\mathrm{NS}}(\pi) \quad \forall (s,a) \in \mathcal{X}$. Then

$$
\begin{aligned}
\sigma_{\mathcal{R}_s^{\mathrm{KL}}(\pi)}(-\pi_s) &= \max_{r(s,a'):r(s,a') \in \mathcal{R}_{s,a'}^{\mathrm{KL}}(\pi), a' \in \mathcal{A}} \sum_{a \in \mathcal{A}} -r(s,a)\pi_s(a) \\
&= \max_{\substack{r(s,a')+\ln(d(a)):\\ r(s,a') \in \mathcal{R}_{s,a'}^{\mathrm{NS}}(\pi), a' \in \mathcal{A}}} \sum_{a \in \mathcal{A}} -r(s,a)\pi_s(a) \\
&= \max_{\substack{r(s,a'):\\ r(s,a') \in \mathcal{R}_{s,a'}^{\mathrm{NS}}(\pi), a' \in \mathcal{A}}} \sum_{a \in \mathcal{A}} -(r(s,a)+\ln(d(a)))\pi_s(a) \\
&= \max_{\substack{r(s,a'):\\ r(s,a') \in \mathcal{R}_{s,a'}^{\mathrm{NS}}(\pi), a' \in \mathcal{A}}} \left\{ -\sum_{a \in \mathcal{A}} \pi_s(a) r(s,a) \right\} - \sum_{a \in \mathcal{A}} \pi_s(a) \ln(d(a)) \\
&= \sum_{a \in \mathcal{A}} \pi_s(a) \ln(\pi_s(a)) - \sum_{a \in \mathcal{A}} \pi_s(a) \ln(d(a)),
\end{aligned}
$$

where the last equality uses Eq. (7). We thus recover the KL divergence $\Omega(\pi_s) = \sum_{a \in \mathcal{A}} \pi_s(a) \ln(\pi_s(a)/d(a))$.

*Negative Tsallis entropy.* Given $\mathcal{R}_{s,a}^{\mathrm{T}}(\pi) := \left[ \frac{1-\pi_s(a)}{2}, +\infty \right) \quad \forall (s,a) \in \mathcal{X}$, we compute:

$$
\begin{aligned}
\sigma_{\mathcal{R}_s^{\mathrm{T}}(\pi)}(-\pi_s) &= \max_{r(s,a'):r(s,a') \in \mathcal{R}_{s,a'}^{\mathrm{T}}(\pi), a' \in \mathcal{A}} \sum_{a \in \mathcal{A}} -r(s,a)\pi_s(a) \\
&= \max_{r(s,a'):r(s,a') \in \left[ \frac{1-\pi_s(a')}{2}, +\infty \right), a' \in \mathcal{A}} \sum_{a \in \mathcal{A}} -r(s,a)\pi_s(a) \\
&= \sum_{a \in \mathcal{A}} -\frac{1-\pi_s(a)}{2}\pi_s(a) \\
&= -\frac{1}{2}\sum_{a \in \mathcal{A}} \pi_s(a) + \frac{1}{2}\sum_{a \in \mathcal{A}} \pi_s(a)^2 = -\frac{1}{2} + \frac{1}{2}\|\pi_s\|^2,
\end{aligned}
\tag{8}
$$

where Eq. (8) also comes from the fact that $\pi_s \geq 0$, and $-r(s,a)\pi_s(a)$ is maximal when $r(s,a)$ is minimal. We thus obtain the negative Tsallis entropy $\Omega(\pi_s) = \frac{1}{2}(\|\pi_s\|^2 - 1)$.

The reward uncertainty sets associated to both KL and Shannon entropy are similar, as the former amounts to translating the latter by a negative constant (translation to the left). As such, both yield reward values that can be either positive or negative. This is not the case of the negative Tsallis, as its minimal reward is 0, attained for a deterministic action policy, *i.e.*, when $\pi_s(a) = 1$.

Table 2 summarizes the properties of each regularizer. For the Tsallis entropy, we denote by $\tau : \mathbb{R}^{\mathcal{A}} \to \mathbb{R}$ the function $q_s \mapsto \frac{\sum_{a \in \mathfrak{A}(q_s)} q_s(a) - 1}{|\mathfrak{A}(q_s)|}$, where $\mathfrak{A}(q_s) \subseteq \mathcal{A}$ is a subset of actions: $\mathfrak{A}(q_s) = \{a \in \mathcal{A} : 1 + i q_s(a_{(i)}) > \sum_{j=0}^{i} q_s(a_{(j)}), i \in \{1, \cdots, |\mathcal{A}|\}\}$, and $a_{(i)}$ is the action with the $i$-th maximal value [20].

## A.5 Proof of Proposition 3.2

**Proposition.** *Assume that $\mathcal{U} = \{P_0\} \times (r_0 + \mathcal{R})$ with $\mathcal{R}_s = \{r_s \in \mathbb{R}^{\mathcal{A}} : \|r_s\| \le \alpha_s^r\}$. Then, the gradient of the reward-robust objective $J_{\mathcal{U}}(\pi) := \langle v^{\pi,\mathcal{U}}, \mu_0 \rangle$ is given by*

$$\nabla J_{\mathcal{U}}(\pi) = \mathbb{E}_{(s,a)\sim\mu_\pi}\left[\nabla \ln \pi_s(a)\left(q^{\pi,\mathcal{U}}(s,a) - \alpha_s^r \frac{\pi_s(a)}{\|\pi_s\|}\right)\right],$$

*where $\mu_\pi$ is the occupancy measure under the nominal model $P_0$ and policy $\pi$.*

We prove the following more general result. To establish Prop. 3.2, we then set $\alpha_s^P = 0$ and apply Thm. 4.1 to replace $v^{\pi,\mathrm{R}^2} = v_{\pi,\mathcal{U}}$ and $q^{\pi,\mathrm{R}^2} = q^{\pi,\mathcal{U}}$.

**Theorem.** *Set $\Omega_{v,\mathrm{R}^2}(\pi_s) := \|\pi_s\|(\alpha_s^r + \alpha_s^P \gamma \|v\|)$. Then, the gradient of the $R^2$ objective $J_{\mathrm{R}^2}(\pi) := \langle v_{\pi,\mathrm{R}^2}, \mu_0 \rangle$ is given by*

$$\nabla J_{\mathrm{R}^2}(\pi) = \mathbb{E}_{s\sim d_{\mu_0,\pi}}\left[\sum_{a\in\mathcal{A}} \pi_s(a)\nabla \ln \pi_s(a) q^{\pi,\mathrm{R}^2}(s,a) - \nabla\Omega_{v,\mathrm{R}^2}(\pi_s)\right],$$

*where $d_{\mu_0,\pi} := \mu_0^\top(\mathbf{I}_{\mathcal{S}} - \gamma P_0^\pi)^{-1}$, with $\mu_0 \in \mathbb{R}^{\mathcal{S}\times 1}$ the initial state distribution.*

*Proof.* By linearity of the gradient operator, $\nabla J_{\mathrm{R}^2}(\pi) = \langle \nabla v^{\pi,\mathrm{R}^2}, \mu_0 \rangle$. We thus need to compute $\nabla v^{\pi,\mathrm{R}^2}$. Using the fixed point property of $v^{\pi,\mathrm{R}^2}$ w.r.t. the $\mathrm{R}^2$ Bellman operator yields:

$$\nabla v^{\pi,\mathrm{R}^2}(s)$$

$$= \nabla\left(r_0^\pi(s) + \gamma P_0^\pi v^{\pi,\mathrm{R}^2}(s) - \Omega_{v,\mathrm{R}^2}(\pi_s)\right)$$

$$= \nabla\left(\sum_{a\in\mathcal{A}} \pi_s(a)(r_0(s,a) + \gamma\langle P_0(\cdot|s,a), v^{\pi,\mathrm{R}^2}\rangle) - \Omega_{v,\mathrm{R}^2}(\pi_s)\right)$$

$$= \sum_{a\in\mathcal{A}} \nabla\pi_s(a)\left(r_0(s,a) + \gamma\langle P_0(\cdot|s,a), v^{\pi,\mathrm{R}^2}\rangle\right)$$

$$\qquad + \gamma\sum_{a\in\mathcal{A}} \pi_s(a)\langle P_0(\cdot|s,a), \nabla v^{\pi,\mathrm{R}^2}\rangle - \nabla\Omega_{v,\mathrm{R}^2}(\pi_s) \qquad\qquad \text{[Linearity of gradient and product rule]}$$

$$= \sum_{a\in\mathcal{A}} \nabla\pi_s(a)q^{\pi,\mathrm{R}^2}(s,a) + \gamma\sum_{a\in\mathcal{A}} \pi_s(a)\langle P_0(\cdot|s,a), \nabla v^{\pi,\mathrm{R}^2}\rangle - \nabla\Omega_{v,\mathrm{R}^2}(\pi_s) \quad [q^{\pi,\mathrm{R}^2}(s,a) = r_0(s,a) + \gamma\langle P_0(\cdot|s,a), v^{\pi,\mathrm{R}^2}\rangle]$$

$$= \sum_{a\in\mathcal{A}} \pi_s(a)(\nabla \ln \pi_s(a)q^{\pi,\mathrm{R}^2}(s,a) + \gamma\langle P_0(\cdot|s,a), \nabla v^{\pi,\mathrm{R}^2}\rangle) - \nabla\Omega_{v,\mathrm{R}^2}(\pi_s) \qquad\qquad [\nabla\pi_s = \pi_s\nabla\ln(\pi_s)]$$

$$= \sum_{a\in\mathcal{A}} \pi_s(a)(\nabla \ln \pi_s(a)q^{\pi,\mathrm{R}^2}(s,a) - \nabla\Omega_{v,\mathrm{R}^2}(\pi_s) + \gamma\langle P_0(\cdot|s,a), \nabla v^{\pi,\mathrm{R}^2}\rangle).$$

Thus, the components of $\nabla v^{\pi,\mathrm{R}^2}$ are the non-regularized value functions corresponding to the modified reward $R(s,a) := \nabla \ln \pi_s(a)q^{\pi,\mathrm{R}^2}(s,a) - \nabla\Omega_{v,\mathrm{R}^2}(\pi_s)$. By the fixed point property of the standard Bellman operator, it results that:

$$\nabla v^{\pi,\mathrm{R}^2}(s) = (\mathbf{I}_{\mathcal{S}} - \gamma P_0^\pi)^{-1}\left(\sum_{a\in\mathcal{A}} \pi.(a)(\nabla \ln \pi.(a)q^{\pi,\mathrm{R}^2}(\cdot,a) - \nabla\Omega_{v,\mathrm{R}^2}(\pi.))\right)(s)$$

and

$$\nabla J_{\mathrm{R}^2}(\pi) = \sum_{s\in\mathcal{S}} \mu_0(s)\nabla v^{\pi,\mathrm{R}^2}(s)$$

$$= \sum_{s\in\mathcal{S}} \mu_0(s)(\mathbf{I}_{\mathcal{S}} - \gamma P_0^\pi)^{-1}\left(\sum_{a\in\mathcal{A}} \pi.(a)(\nabla \ln \pi.(a)q^{\pi,\mathrm{R}^2}(\cdot,a) - \nabla\Omega_{v,\mathrm{R}^2}(\pi.))\right)(s)$$

$$= \sum_{s\in\mathcal{S}} d_{\mu_0,\pi}(s)\left(\sum_{a\in\mathcal{A}} \pi_s(a)\nabla \ln \pi_s(a)q^{\pi,\mathrm{R}^2}(s,a) - \nabla\Omega_{v,\mathrm{R}^2}(\pi_s)\right),$$

by definition of $d_{\mu_0,\pi}$. $\qquad\square$

The subtraction by $\nabla\Omega_{v,\mathrm{R}^2}(\pi_s)$ also appears in [10]. However, here, the gradient includes partial derivatives that depend on both the policy and the value itself. Let's try to compute the gradient of the double regularizer $\Omega_{v,\mathrm{R}^2}(\pi_s) = \|\pi_s\|(\alpha_s^r + \alpha_s^P\gamma\|v^{\pi,\mathrm{R}^2}\|)$. By the chain-rule we have that:

$$\nabla\Omega_{v,\mathrm{R}^2}(\pi_s) = \sum_{a\in\mathcal{A}}\frac{\partial\Omega_{v,\mathrm{R}^2}}{\partial\pi_s(a)}\nabla\pi_s(a) + \sum_{s\in\mathcal{S}}\frac{\partial\Omega_{v,\mathrm{R}^2}}{\partial v^{\pi,\mathrm{R}^2}(s)}\nabla v^{\pi,\mathrm{R}^2}(s)$$

$$= \sum_{a\in\mathcal{A}}(\alpha_s^r + \alpha_s^P\gamma\|v^{\pi,\mathrm{R}^2}\|)\frac{\pi_s(a)}{\|\pi_s\|}\nabla\pi_s(a) + \sum_{s\in\mathcal{S}}\alpha_s^P\gamma\|\pi_s\|\frac{v^{\pi,\mathrm{R}^2}(s)}{\|v^{\pi,\mathrm{R}^2}\|}\nabla v^{\pi,\mathrm{R}^2}(s)$$

$$= \sum_{a\in\mathcal{A}}\pi_s(a)\left(\frac{\alpha_s^r + \alpha_s^P\gamma\|v^{\pi,\mathrm{R}^2}\|}{\|\pi_s\|}\nabla\pi_s(a) + \sum_{s\in\mathcal{S}}\alpha_s^P\gamma\|\pi_s\|\frac{q^{\pi,\mathrm{R}^2}(s,a)}{\|v^{\pi,\mathrm{R}^2}\|}\nabla v^{\pi,\mathrm{R}^2}(s)\right).$$

We remark here an interdependence between $\nabla\Omega_{v,\mathrm{R}^2}(\pi_s)$ and $\nabla v^{\pi,\mathrm{R}^2}(s)$: computing the gradient $\nabla\Omega_{v,\mathrm{R}^2}(\pi_s)$ requires to know $\nabla v^{\pi,\mathrm{R}^2}(s)$ and vice versa. There may be a recursion that still enables to compute these gradients, which we leave for future work.

# B General robust MDPs

## B.1 Proof of Theorem 4.1

**Theorem** (General robust MDP). *Assume that $\mathcal{U} = (P_0 + \mathcal{P}) \times (r_0 + \mathcal{R})$. Then, for any policy $\pi \in \Delta_{\mathcal{A}}^{\mathcal{S}}$, the robust value function $v^{\pi,\mathcal{U}}$ is the optimal solution of the convex optimization problem:*

$$\max_{v\in\mathbb{R}^{\mathcal{S}}}\langle v,\mu_0\rangle \text{ s. t. } v(s) \leq T_{(P_0,r_0)}^{\pi}v(s) - \sigma_{\mathcal{R}_s}(-\pi_s) - \sigma_{\mathcal{P}_s}(-\gamma v\cdot\pi_s)\text{ for all } s\in\mathcal{S},$$

*where $[v\cdot\pi_s](s',a) := v(s')\pi_s(a) \quad \forall(s',a)\in\mathcal{X}$.*

*Proof.* The robust value function $v^{\pi,\mathcal{U}}$ is the optimal solution of:

$$\max_{v\in\mathbb{R}^{\mathcal{S}}}\langle v,\mu_0\rangle \text{ s. t. } F(s)\leq 0 \text{ for all } s\in\mathcal{S}, \tag{9}$$

where $F(s) := \max_{(P,r)\in\mathcal{U}}\{v(s) - r^{\pi}(s) - \gamma P^{\pi}v(s)\}$ is the robust counterpart of $(\mathrm{P}_{\mathcal{U}})$ at $s\in\mathcal{S}$. Let's compute it based on the structure of the uncertainty set $\mathcal{U} = (P_0+\mathcal{P})\times(r_0+\mathcal{R})$:

$$F(s) = \max_{(P',r')\in(P_0+\mathcal{P})\times(r_0+\mathcal{R})}\{v(s) - r'^{\pi}(s) - \gamma P'^{\pi}v(s)\}$$

$$= \max_{\substack{P':P'=P_0+P,P\in\mathcal{P}\\r':r'=r_0+r,r\in\mathcal{R}}}\{v(s) - r'^{\pi}(s) - \gamma P'^{\pi}v(s)\}$$

$$= \max_{P\in\mathcal{P},r\in\mathcal{R}}\{v(s) - (r_0^{\pi}(s) + r^{\pi}(s)) - \gamma(P_0^{\pi} + P^{\pi})v(s)\} \qquad [(P_0+P)^{\pi} = P_0^{\pi} + P^{\pi},$$

$$(r_0+r)^{\pi} = r_0^{\pi} + r^{\pi}]$$

$$= \max_{P\in\mathcal{P},r\in\mathcal{R}}\{v(s) - r_0^{\pi}(s) - r^{\pi}(s) - \gamma P_0^{\pi}v(s) - \gamma P^{\pi}v(s)\}$$

$$= \max_{P\in\mathcal{P},r\in\mathcal{R}}\left\{v(s) - T_{(P_0,r_0)}^{\pi}v(s) - r^{\pi}(s) - \gamma P^{\pi}v(s)\right\} \qquad [T_{(P_0,r_0)}^{\pi}v(s) = r_0^{\pi}(s) + \gamma P_0^{\pi}v(s)]$$

$$= \max_{P\in\mathcal{P}}\{-\gamma P^{\pi}v(s)\} + \max_{r\in\mathcal{R}}\{-r^{\pi}(s)\} + v(s) - T_{(P_0,r_0)}^{\pi}v(s)$$

$$= -\min_{P\in\mathcal{P}}\{\gamma P^{\pi}v(s)\} - \min_{r\in\mathcal{R}}\{r^{\pi}(s)\} + v(s) - T_{(P_0,r_0)}^{\pi}v(s)$$

$$= -\min_{P\in\mathbb{R}^{\mathcal{X}\times\mathcal{S}}}\{\gamma P^{\pi}v(s) + \delta_{\mathcal{P}}(P)\} - \min_{r\in\mathbb{R}^{\mathcal{X}}}\{r^{\pi}(s) + \delta_{\mathcal{R}}(r)\}$$

$$+ v(s) - T_{(P_0,r_0)}^{\pi}v(s)$$

$$= -\min_{P\in\mathbb{R}^{\mathcal{X}\times\mathcal{S}}}\{\gamma\langle P_s^{\pi},v\rangle + \delta_{\mathcal{P}}(P)\} - \min_{r\in\mathbb{R}^{\mathcal{X}}}\{\langle r_s,\pi_s\rangle + \delta_{\mathcal{R}}(r)\}$$

$$+ v(s) - T_{(P_0,r_0)}^{\pi}v(s). \qquad [P^{\pi}v(s) = \langle P_s^{\pi},v\rangle, r^{\pi}(s) = \langle r_s,\pi_s\rangle]$$

As shown in the proof of Thm. 3.1, $\min_{r\in\mathbb{R}^{\mathcal{X}}}\{\langle r_s, \pi_s\rangle + \delta_{\mathcal{R}}(r)\} = \min_{r_s\in\mathbb{R}^{\mathcal{A}}}\{\langle r_s, \pi_s\rangle + \delta_{\mathcal{R}_s}(r_s)\}$ thanks to the rectangularity assumption. Similarly, by rectangularity of the transition uncertainty set, for all $P := (P_s)_{s\in\mathcal{S}} \in \mathbb{R}^{\mathcal{X}}$, we have $\delta_{\mathcal{P}}(P) = \sum_{s'\in\mathcal{S}} \delta_{\mathcal{P}_{s'}}(P_{s'})$. As such,

$$\min_{P\in\mathbb{R}^{\mathcal{X}\times\mathcal{S}}}\{\gamma\langle P_s^{\pi}, v\rangle + \delta_{\mathcal{P}}(P)\} = \min_{P\in\mathbb{R}^{\mathcal{X}\times\mathcal{S}}}\{\gamma\langle P_s^{\pi}, v\rangle + \sum_{s'\in\mathcal{S}} \delta_{\mathcal{P}_{s'}}(P_{s'})\}$$
$$= \min_{P_s\in\mathbb{R}^{\mathcal{X}}}\{\gamma\langle P_s^{\pi}, v\rangle + \delta_{\mathcal{P}_s}(P_s)\},$$

where the last equality holds since the objective function is minimal if and only if $P_s \in \mathcal{P}_s$. Finally,

$$F(s) = -\min_{P_s\in\mathbb{R}^{\mathcal{X}}}\{\gamma\langle P_s^{\pi}, v\rangle + \delta_{\mathcal{P}_s}(P_s)\} - \min_{r_s\in\mathbb{R}^{\mathcal{A}}}\{\langle r_s, \pi_s\rangle + \delta_{\mathcal{R}_s}(r_s)\} + v(s) - T_{(P_0, r_0)}^{\pi}v(s).$$

Referring to the proof of Thm. 3.1, we know that $-\min_{r\in\mathbb{R}^{\mathcal{X}}}\{\langle r_s, \pi_s\rangle + \delta_{\mathcal{R}}(r)\} = \sigma_{\mathcal{R}_s}(-\pi_s)$, so

$$F(s) = -\min_{P_s\in\mathbb{R}^{\mathcal{X}}}\{\gamma\langle P_s^{\pi}, v\rangle + \delta_{\mathcal{P}_s}(P_s)\} + \sigma_{\mathcal{R}_s}(-\pi_s) + v(s) - T_{(P_0, r_0)}^{\pi}v(s).$$

Let the matrix $v \cdot \pi_s \in \mathbb{R}^{\mathcal{X}}$ defined as $[v \cdot \pi_s](s', a) := v(s')\pi_s(a)$ for all $(s', a) \in \mathcal{X}$. Further define $\varphi(P_s) := \gamma\langle P_s^{\pi}, v\rangle$, which we can rewrite as $\varphi(P_s) = \gamma\langle P_s, v \cdot \pi_s\rangle$. Then, we have that:

$$\min_{P_s\in\mathbb{R}^{\mathcal{X}}}\{\gamma\langle P_s^{\pi}, v\rangle + \delta_{\mathcal{P}_s}(P_s)\} = \min_{P_s\in\mathbb{R}^{\mathcal{X}}}\{\varphi(P_s) + \delta_{\mathcal{P}_s}(P_s)\} = -\min_{\mathbf{B}\in\mathbb{R}^{\mathcal{X}}}\{\varphi^*(-\mathbf{B}) + \sigma_{\mathcal{P}_s}(\mathbf{B})\},$$

where the last equality results from Fenchel-Rockafellar duality and the fact that $(\delta_{\mathcal{P}_s})^* = \sigma_{\mathcal{P}_s}$. It thus remains to compute the convex conjugate of $\varphi$:

$$\varphi^*(-\mathbf{B}) = \max_{P_s\in\mathbb{R}^{\mathcal{X}}}\{\langle P_s, -\mathbf{B}\rangle - \varphi(P_s)\}$$
$$= \max_{P_s\in\mathbb{R}^{\mathcal{X}}}\{\langle P_s, -\mathbf{B}\rangle - \gamma\langle P_s, v \cdot \pi_s\rangle\}$$
$$= \max_{P_s\in\mathbb{R}^{\mathcal{X}}}\langle P_s, -\mathbf{B} -\gamma v \cdot \pi_s\rangle$$
$$= \begin{cases} 0 \text{ if } -\mathbf{B} -\gamma v \cdot \pi_s = 0 \\ +\infty \text{ otherwise,} \end{cases}$$

which yields $\min_{\mathbf{B}\in\mathbb{R}^{\mathcal{X}}}\{\varphi^*(-\mathbf{B}) + \sigma_{\mathcal{P}_s}(\mathbf{B})\} = \sigma_{\mathcal{P}_s}(-\gamma v \cdot \pi_s)$. Finally, the robust counterpart rewrites as: $F(s) = \sigma_{\mathcal{P}_s}(-\gamma v \cdot \pi_s) + \sigma_{\mathcal{R}_s}(-\pi_s) + v(s) - T_{(P_0, r_0)}^{\pi}v(s)$, and plugging it into the optimization problem (9) yields the desired result. $\qquad\square$

### B.2 Proof of Corollary 4.1

**Corollary.** *Assume that* $\mathcal{U} = (P_0 + \mathcal{P}) \times (r_0 + \mathcal{R})$ *with* $\mathcal{P}_s := \{P_s \in \mathbb{R}^{\mathcal{X}} : \|P_s\| \leq \alpha_s^P\}$ *and* $\mathcal{R}_s := \{r_s \in \mathbb{R}^{\mathcal{A}} : \|r_s\| \leq \alpha_s^r\}$ *for all* $s \in \mathcal{S}$. *Then, the robust value function* $v^{\pi, \mathcal{U}}$ *is the optimal solution of the convex optimization problem:*

$$\max_{v\in\mathbb{R}^{\mathcal{S}}}\langle v, \mu_0\rangle \text{ s. t. } v(s) \leq T_{(P_0, r_0)}^{\pi}v(s) - \alpha_s^r\|\pi_s\| - \alpha_s^P\gamma\|v\|\|\pi_s\| \text{ for all } s \in \mathcal{S}.$$

*Proof.* As we already showed in Cor. 3.1, the support function of the reward uncertainty set is $\sigma_{\mathcal{R}_s}(-\pi_s) = \alpha_s^r\|\pi_s\|$. For the transition uncertainty set, we similarly have:

$$\sigma_{\mathcal{P}_s}(-\gamma v \cdot \pi_s) = \max_{\substack{P_s\in\mathbb{R}^{\mathcal{X}}: \\ \|P_s\|\leq\alpha_s^P}} \langle P_s, -\gamma v \cdot \pi_s\rangle$$
$$= \alpha_s^P\|-\gamma v \cdot \pi_s\|$$
$$= \alpha_s^P\gamma\|v \cdot \pi_s\|$$
$$= \alpha_s^P\gamma\|v\|\|\pi_s\|. \qquad\qquad [\|v \cdot \pi_s\|^2 = \sum_{(s', a)\in\mathcal{X}} (v(s')\pi_s(a))^2$$
$$= \sum_{s'\in\mathcal{S}} v(s')^2 \sum_{a\in\mathcal{A}} \pi_s(a)^2 = \|v\|^2\|\pi_s\|^2]$$

Now we apply Thm. 3.1 and replace each support function by their explicit form to get that the robust value function $v^{\pi,\mathcal{U}}$ is the optimal solution of:

$$\max_{v\in\mathbb{R}^{\mathcal{S}}}\langle v,\mu_0\rangle \text{ s. t. } v(s)\le T^{\pi}_{(P_0,r_0)}v(s)-\alpha^r_s\|\pi_s\|-\alpha^P_s\|\pi_s\|\cdot\gamma\|v\| \text{ for all } s\in\mathcal{S}.$$

*Ball-constraints with arbitrary norms.* As seen in the proof of Thm. 3.1 and Cor. 3.1, for ball-constrained rewards defined with an arbitrary norm $\|\cdot\|_a$ of dual $\|\cdot\|_{a*}$, the corresponding support function is $\sigma_{\mathcal{R}_s}(-\pi_s)=\alpha^r_s\|\pi_s\|_{a*}$. Similarly, for ball-constrained transitions based on a norm $\|\cdot\|_b$ of dual $\|\cdot\|_{b*}$, we have:

$$\sigma_{\mathcal{P}_s}(-\gamma v\cdot\pi_s)=\max_{\substack{P_s\in\mathbb{R}^{\mathcal{X}}:\\ \|P_s\|_b\le\alpha^P_s}}\langle P_s,-\gamma v\cdot\pi_s\rangle=\alpha^P_s\|-\gamma v\cdot\pi_s\|_{b*},$$

in which case the robust value function $v^{\pi,\mathcal{U}}$ is the optimal solution of:

$$\max_{v\in\mathbb{R}^{\mathcal{S}}}\langle v,\mu_0\rangle \text{ s. t. } v(s)\le T^{\pi}_{(P_0,r_0)}v(s)-\alpha^r_s\|\pi_s\|_{a*}-\alpha^P_s\|-\gamma v\cdot\pi_s\|_{b*} \text{ for all } s\in\mathcal{S}.$$

$\square$

# C $\quad$ R$^2$ MDPs

## C.1 $\quad$ Proof of Proposition 5.1

**Proposition.** *Suppose that Asm. 5.1 holds. Then, we have the following properties:*
*(i) Monotonicity: For all $v_1,v_2\in\mathbb{R}^{\mathcal{S}}$ such that $v_1\le v_2$, we have $T^{\pi,\mathrm{R}^2}v_1\le T^{\pi,\mathrm{R}^2}v_2$ and $T^{*,\mathrm{R}^2}v_1\le T^{*,\mathrm{R}^2}v_2$.*
*(ii) Sub-distributivity: For all $v_1\in\mathbb{R}^{\mathcal{S}},c\in\mathbb{R}$, we have $T^{\pi,\mathrm{R}^2}(v_1+c\mathbb{1}_{\mathcal{S}})\le T^{\pi,\mathrm{R}^2}v_1+\gamma c\mathbb{1}_{\mathcal{S}}$ and $T^{*,\mathrm{R}^2}(v_1+c\mathbb{1}_{\mathcal{S}})\le T^{*,\mathrm{R}^2}v_1+\gamma c\mathbb{1}_{\mathcal{S}},\forall c\in\mathbb{R}$.*
*(iii) Contraction: Let $\epsilon_*:=\min_{s\in\mathcal{S}}\epsilon_s>0$. Then, for all $v_1,v_2\in\mathbb{R}^{\mathcal{S}}$, $\|T^{\pi,\mathrm{R}^2}v_1-T^{\pi,\mathrm{R}^2}v_2\|_\infty\le(1-\epsilon_*)\|v_1-v_2\|_\infty$ and $\|T^{*,\mathrm{R}^2}v_1-T^{*,\mathrm{R}^2}v_2\|_\infty\le(1-\epsilon_*)\|v_1-v_2\|_\infty$.*

*Proof. Proof of (i).* Consider the evaluation operator and let $v_1,v_2\in\mathbb{R}^{\mathcal{S}}$ such that $v_1\le v_2$. For all $s\in\mathcal{S}$,

$$[T^{\pi,\mathrm{R}^2}v_1-T^{\pi,\mathrm{R}^2}v_2](s)$$
$$= T^{\pi}_{(P_0,r_0)}v_1(s)-\alpha^r_s\|\pi_s\|-\alpha^P_s\gamma\|v_1\|\|\pi_s\|$$
$$\quad -(T^{\pi}_{(P_0,r_0)}v_2(s)-\alpha^r_s\|\pi_s\|-\alpha^P_s\gamma\|v_2\|\|\pi_s\|)$$
$$= T^{\pi}_{(P_0,r_0)}v_1(s)-T^{\pi}_{(P_0,r_0)}v_2(s)+\alpha^P_s\gamma\|\pi_s\|(\|v_2\|-\|v_1\|)$$
$$= \gamma P^{\pi}_0(v_1-v_2)(s)+\alpha^P_s\gamma\|\pi_s\|(\|v_2\|-\|v_1\|)$$
$$= \gamma\langle\pi_s,P_{0s}(v_1-v_2)\rangle+\alpha^P_s\gamma\|\pi_s\|(\|v_2\|-\|v_1\|) \qquad [\forall v\in\mathbb{R}^{\mathcal{S}},P^{\pi}_0v(s)=\sum_{(s',a)\in\mathcal{X}}\pi_s(a)P_0(s'|s,a)v(s')$$
$$\qquad\qquad\qquad\qquad\qquad\qquad\qquad\qquad\qquad\qquad =\sum_{a\in\mathcal{A}}\pi_s(a)[P_{0s}v](a)=\langle\pi_s,P_{0s}v\rangle]$$
$$= \gamma\|\pi_s\|\left(\left\langle\frac{\pi_s}{\|\pi_s\|},P_{0s}(v_1-v_2)\right\rangle+\alpha^P_s(\|v_2\|-\|v_1\|)\right)$$
$$\le \gamma\|\pi_s\|\left(\left\langle\frac{\pi_s}{\|\pi_s\|},P_{0s}(v_1-v_2)\right\rangle+\alpha^P_s(\|v_2-v_1\|)\right) \qquad [\forall v,w\in\mathbb{R}^{\mathcal{S}},\|v\|-\|w\|\le|\|v\|-\|w\||\le\|v-w\|].$$

By Asm. 5.1, we also have

$$\alpha^P_s\le\min_{\substack{\mathbf{u}_{\mathcal{A}}\in\mathbb{R}^{\mathcal{A}}_+,\|\mathbf{u}_{\mathcal{A}}\|=1\\ \mathbf{v}_{\mathcal{S}}\in\mathbb{R}^{\mathcal{S}}_+,\|\mathbf{v}_{\mathcal{S}}\|=1}}\mathbf{u}^{\top}_{\mathcal{A}}P_0(\cdot|s,\cdot)\mathbf{v}_{\mathcal{S}}=\min_{\substack{\mathbf{u}_{\mathcal{A}}\in\mathbb{R}^{\mathcal{A}}_+,\|\mathbf{u}_{\mathcal{A}}\|=1\\ \mathbf{v}_{\mathcal{S}}\in\mathbb{R}^{\mathcal{S}}_+,\|\mathbf{v}_{\mathcal{S}}\|=1}}\langle\mathbf{u}_{\mathcal{A}},P_0(\cdot|s,\cdot)\mathbf{v}_{\mathcal{S}}\rangle\le\left\langle\frac{\pi_s}{\|\pi_s\|},P_0(\cdot|s,\cdot)\frac{(v_2-v_1)}{\|v_2-v_1\|}\right\rangle,$$

so that

$$[T^{\pi,\mathrm{R}^2}v_1 - T^{\pi,\mathrm{R}^2}v_2](s) \le \gamma\|\pi_s\|\left(\left\langle\frac{\pi_s}{\|\pi_s\|}, P_{0s}(v_1-v_2)\right\rangle + \left\langle\frac{\pi_s}{\|\pi_s\|}, P_0(\cdot|s,\cdot)\frac{(v_2-v_1)}{\|v_2-v_1\|}\right\rangle\|v_2-v_1\|\right)$$

$$= \gamma\|\pi_s\|\left(\left\langle\frac{\pi_s}{\|\pi_s\|}, P_{0s}(v_1-v_2)\right\rangle + \left\langle\frac{\pi_s}{\|\pi_s\|}, P_0(\cdot|s,\cdot)(v_2-v_1)\right\rangle\right) = 0,$$

where we switch notations to designate $P_0(\cdot|s,\cdot) = P_{0s} \in \mathbb{R}^{\mathcal{A}\times\mathcal{S}}$. This proves monotonicity.

*Proof of (ii).* We now prove the sub-distributivity of the evaluation operator. Let $v \in \mathbb{R}^{\mathcal{S}}, c \in \mathbb{R}$. For all $s \in \mathcal{S}$,

$$\begin{aligned}
&[T^{\pi,\mathrm{R}^2}(v + c\mathbb{1}_{\mathcal{S}})](s)\\
=&[T^\pi_{(P_0,r_0)}(v + c\mathbb{1}_{\mathcal{S}})](s) - \alpha_s^r\|\pi_s\| - \alpha_s^P\gamma\|v + c\mathbb{1}_{\mathcal{S}}\|\|\pi_s\|\\
=&T^\pi_{(P_0,r_0)}v(s) + \gamma c - \alpha_s^r\|\pi_s\| - \alpha_s^P\gamma\|v + c\mathbb{1}_{\mathcal{S}}\|\|\pi_s\| \qquad [T^\pi_{(P_0,r_0)}(v + c\mathbb{1}_{\mathcal{S}}) = T^\pi_{(P_0,r_0)}v + \gamma c\mathbb{1}_{\mathcal{S}}]\\
\le&T^\pi_{(P_0,r_0)}v(s) + \gamma c - \alpha_s^r\|\pi_s\| - \alpha_s^P\gamma\|\pi_s\|(\|v\| + \|c\mathbb{1}_{\mathcal{S}}\|)\\
=&T^\pi_{(P_0,r_0)}v(s) + \gamma c - \alpha_s^r\|\pi_s\| - \alpha_s^P\gamma\|\pi_s\|\|v\|\\
&\quad - \alpha_s^P\gamma\|\pi_s\|\|c\mathbb{1}_{\mathcal{S}}\|\\
=&[T^{\pi,\mathrm{R}^2}v](s) + \gamma c - \alpha_s^P\gamma\|\pi_s\|\|c\mathbb{1}_{\mathcal{S}}\|\\
\le&[T^{\pi,\mathrm{R}^2}v](s) + \gamma c. \qquad\qquad\qquad\qquad\qquad\qquad\qquad [\gamma > 0, \alpha_s^P > 0, \|\cdot\| \ge 0]
\end{aligned}$$

*Proof of (iii).* We prove the contraction of a more general evaluation operator with $\ell_p$ regularization, $p \ge 1$. This will establish contraction of the R$^2$ operator $T^{\pi,\mathrm{R}^2}$ by simply setting $p = 2$. Define as $q$ the conjugate value of $p$, *i.e.*, such that $\frac{1}{p} + \frac{1}{q} = 1$. As seen in the proof of Thm. 4.1, for balls that are constrained according to the $\ell_p$-norm $\|\cdot\|_p$, the robust value function $v^{\pi,\mathcal{U}}$ is the optimal solution of:

$$\max_{v\in\mathbb{R}^{\mathcal{S}}}\langle v, \mu_0\rangle \text{ s. t. } v(s) \le T^\pi_{(P_0,r_0)}v(s) - \alpha_s^r\|\pi_s\|_q - \alpha_s^P\|-\gamma v\cdot\pi_s\|_q \text{ for all } s \in \mathcal{S},$$

because $\|\cdot\|_q$ is the dual norm of $\|\cdot\|_p$, and we can define the R$^2$ operator accordingly:

$$[T_q^{\pi,\mathrm{R}^2}v](s) := T^\pi_{(P_0,r_0)}v(s) - \alpha_s^r\|\pi_s\|_q - \alpha_s^P\gamma\|v\cdot\pi_s\|_q \quad \forall v \in \mathbb{R}^{\mathcal{S}}, s \in \mathcal{S}.$$

We make the following assumption:

**Assumption** ($\mathrm{A}_q$). *For all $s \in \mathcal{S}$, there exists $\epsilon_s > 0$ such that $\alpha_s^P \le \frac{1-\gamma-\epsilon_s}{\gamma|\mathcal{S}|^{\frac{1}{q}}}$.*

Let $v_1, v_2 \in \mathbb{R}^{\mathcal{S}}$. For all $s \in \mathcal{S}$,

$$
\left| [T_q^{\pi,\mathrm{R}^2} v_1](s) - [T_q^{\pi,\mathrm{R}^2} v_2](s) \right|
$$

$$
= |\ T_{(P_0,r_0)}^{\pi} v_1(s) - \alpha_s^r \|\pi_s\|_q - \alpha_s^P \gamma \|v_1 \cdot \pi_s\|_q
$$
$$
\qquad - (T_{(P_0,r_0)}^{\pi} v_2(s) - \alpha_s^r \|\pi_s\|_q - \alpha_s^P \gamma \|v_2 \cdot \pi_s\|_q)\ |
$$

$$
= \left| T_{(P_0,r_0)}^{\pi} v_1(s) - T_{(P_0,r_0)}^{\pi} v_2(s) \right| + \left| \alpha_s^P \gamma (\|v_2 \cdot \pi_s\|_q - \|v_1 \cdot \pi_s\|_q) \right|
$$

$$
= \left| T_{(P_0,r_0)}^{\pi} v_1(s) - T_{(P_0,r_0)}^{\pi} v_2(s) \right| + \alpha_s^P \gamma \left| \|v_2 \cdot \pi_s\|_q - \|v_1 \cdot \pi_s\|_q \right|
$$

$$
\leq \left| T_{(P_0,r_0)}^{\pi} v_1(s) - T_{(P_0,r_0)}^{\pi} v_2(s) \right| + \alpha_s^P \gamma \|v_2 \cdot \pi_s - v_1 \cdot \pi_s\|_q \qquad [\forall \mathbf{A}, \mathbf{B} \in \mathbb{R}^{\mathcal{X}}, |\|\mathbf{A}\|_q - \|\mathbf{B}\|_q| \leq \|\mathbf{A} - \mathbf{B}\|_q]
$$

$$
\leq \gamma \|v_1 - v_2\|_\infty + \alpha_s^P \gamma \|v_2 \cdot \pi_s - v_1 \cdot \pi_s\|_q \qquad [\|T_{(P_0,r_0)}^{\pi} v_1 - T_{(P_0,r_0)}^{\pi} v_2\|_\infty \leq \gamma \|v_1 - v_2\|_\infty]
$$

$$
= \gamma \|v_1 - v_2\|_\infty + \alpha_s^P \gamma \|(v_2 - v_1) \cdot \pi_s\|_q \qquad [\forall v, w \in \mathbb{R}^{\mathcal{S}}, v \cdot \pi_s - w \cdot \pi_s = (v - w) \cdot \pi_s]
$$

$$
\leq \gamma \|v_1 - v_2\|_\infty + \alpha_s^P \gamma \|v_2 - v_1\|_q \qquad [\forall v \in \mathbb{R}^{\mathcal{S}}, \|v \cdot \pi_s\|_q \leq \|v\|_q]
$$

$$
\leq \gamma \|v_1 - v_2\|_\infty + \alpha_s^P \gamma |\mathcal{S}|^{\frac{1}{q}} \|v_1 - v_2\|_\infty \qquad [\forall v, w \in \mathbb{R}^{\mathcal{S}}, \|v - w\|_q \leq |\mathcal{S}|^{\frac{1}{q}} \|v - w\|_\infty]
$$

$$
= \gamma (1 + \alpha_s^P |\mathcal{S}|^{\frac{1}{q}}) \|v_1 - v_2\|_\infty
$$

$$
\leq \gamma \left( 1 + \frac{1 - \gamma - \epsilon_s}{\gamma} \right) \|v_1 - v_2\|_\infty \qquad \left[ \alpha_s^P \leq \frac{1 - \gamma - \epsilon_s}{\gamma |\mathcal{S}|^{\frac{1}{q}}} \text{ by Asm. (A}_q) \right]
$$

$$
= (1 - \epsilon_s) \|v_1 - v_2\|_\infty
$$

$$
\leq (1 - \epsilon_*) \|v_1 - v_2\|_\infty,
$$

where $\epsilon_* := \min_{s \in \mathcal{S}} \epsilon_s$. Setting $q = 2$ and remarking that: (i) the first bound in Asm. 5.1 recovers Asm. (A$_q$); (ii) $T_2^{\pi,\mathrm{R}^2} = T^{\pi,\mathrm{R}^2}$, establishes contraction of the R$^2$ evaluation operator. For the optimality operator, the proof is exactly the same as that of [10, Prop. 3], using Prop. 2.1. $\qquad\square$

## C.2  Proof of Theorem 5.1

**Theorem** (R$^2$ optimal policy). *The greedy policy $\pi^{*,\mathrm{R}^2} = \mathcal{G}_{\Omega_{\mathrm{R}^2}}(v^{*,\mathrm{R}^2})$ is the unique optimal R$^2$ policy, i.e., for all $\pi \in \Delta_{\mathcal{A}}^{\mathcal{S}}, v^{\pi^{*,\mathrm{R}^2}} = v^{*,\mathrm{R}^2} \geq v^{\pi,\mathrm{R}^2}$.*

*Proof.* By strong convexity of the norm, the R$^2$ function $\Omega_{v,\mathrm{R}^2} : \pi_s \mapsto \|\pi_s\|(\alpha_s^r + \alpha_s^P \gamma \|v\|)$ is strongly convex in $\pi_s$. As such, we can invoke Prop. 2.1 to state that the greedy policy $\pi^{*,\mathrm{R}^2}$ is the unique maximizing argument for $v^{*,\mathrm{R}^2}$. Moreover, by construction,

$$
T^{\pi^{*,\mathrm{R}^2},\mathrm{R}^2} v^{*,\mathrm{R}^2} = T^{*,\mathrm{R}^2} v^{*,\mathrm{R}^2} = v^{*,\mathrm{R}^2}.
$$

Supposing that Asm. 5.1 holds, the evaluation operator $T^{\pi^{*,\mathrm{R}^2},\mathrm{R}^2}$ is contracting and has a unique fixed point $v^{\pi^{*,\mathrm{R}^2},\mathrm{R}^2}$. Therefore, $v^{*,\mathrm{R}^2}$ being also a fixed point, we have $v^{\pi^{*,\mathrm{R}^2},\mathrm{R}^2} = v^{*,\mathrm{R}^2}$. It remains to show the last inequality: the proof is exactly the same as that of [10, Thm. 1], and relies on the monotonicity of the R$^2$ operators. $\qquad\square$

## C.3  Proof of Remark 5.1

**Remark C.1.** *An optimal R$^2$ policy may be stochastic. This is due to the fact that our R$^2$ MDP framework builds upon the general $s$-rectangularity assumption. Robust MDPs with $s$-rectangular uncertainty sets similarly yield an optimal robust policy that is stochastic [40, Table 1]. Nonetheless, the R$^2$ MDP formulation recovers a deterministic optimal policy in the more specific $(s,a)$-rectangular case, which is in accordance with the robust MDP setting.*

*Proof.* In the $(s,a)$-rectangular case, the uncertainty set is structured as $\mathcal{U} = \times_{(s,a)\in\mathcal{X}}\mathcal{U}(s,a)$, where $\mathcal{U}(s,a) := P_0(\cdot|s,a) \times r_0(s,a) + \mathcal{P}(s,a) \times \mathcal{R}(s,a)$. The robust counterpart of problem (P$_{\mathcal{U}}$) is:

$$F(s) = \max_{(P,r)\in\mathcal{U}} \{v(s) - r^\pi(s) - \gamma P^\pi v(s)\}$$

$$= \max_{(P(\cdot|s,a),r(s,a))\in\mathcal{P}(s,a)\times\mathcal{R}(s,a)} \{v(s) - r_0^\pi(s) - r^\pi(s) - \gamma P_0^\pi v(s) - \gamma P^\pi v(s)\}$$

$$= \max_{(P(\cdot|s,a),r(s,a))\in\mathcal{P}(s,a)\times\mathcal{R}(s,a)} \{-r^\pi(s) - \gamma P^\pi v(s)\} + v(s) - r_0^\pi(s) - \gamma P_0^\pi v(s)$$

$$= \max_{r(s,a)\in\mathcal{R}(s,a)} \{-r^\pi(s)\} + \gamma \max_{P(\cdot|s,a)\in\mathcal{P}(s,a)} \{-P^\pi v(s)\} + v(s) - T_{(P_0,r_0)}^\pi v(s)$$

$$= \max_{r(s,a)\in\mathcal{R}(s,a)} \left\{-\sum_{a\in\mathcal{A}}\pi_s(a)r(s,a)\right\} + \gamma \max_{P(\cdot|s,a)\in\mathcal{P}(s,a)} \left\{-\sum_{a\in\mathcal{A}}\pi_s(a)\langle P(\cdot|s,a),v\rangle\right\}$$

$$\quad + v(s) - T_{(P_0,r_0)}^\pi v(s)$$

$$= \sum_{a\in\mathcal{A}}\pi_s(a)\left(\max_{r(s,a)\in\mathcal{R}(s,a)}\{-r(s,a)\} + \gamma \max_{P(\cdot|s,a)\in\mathcal{P}(s,a)}\{\langle P(\cdot|s,a),-v\rangle\}\right) + v(s) - T_{(P_0,r_0)}^\pi v(s).$$

In particular, if we have ball uncertainty sets $\mathcal{P}(s,a) := \{P(\cdot|s,a) \in \mathbb{R}^{\mathcal{S}} : \|P(\cdot|s,a)\| \leq \alpha_{s,a}^P\}$ and $\mathcal{R}(s,a) := \{r(s,a) \in \mathbb{R} : |r(s,a)| \leq \alpha_{s,a}^r\}$ for all $(s,a) \in \mathcal{X}$, then we can explicitly compute the support functions:

$$\max_{r(s,a):|r(s,a)|\leq\alpha_{s,a}^r} -r(s,a) = \alpha_{s,a}^r \text{ and } \max_{P(\cdot|s,a):\|P(\cdot|s,a)\|\leq\alpha_{s,a}^P} \langle P(\cdot|s,a),-v\rangle = \alpha_{s,a}^P\|v\|.$$

Therefore, the robust counterpart rewrites as:

$$F(s) = \sum_{a\in\mathcal{A}}\pi_s(a)(\alpha_{s,a}^r + \gamma\alpha_{s,a}^P\|v\|) + v(s) - T_{(P_0,r_0)}^\pi v(s),$$

and the robust value function $v^{\pi,\mathcal{U}}$ is the optimal solution of the convex optimization problem:

$$\max_{v\in\mathbb{R}^{\mathcal{S}}}\langle v,\mu_0\rangle \text{ s. t. } v(s) \leq T_{(P_0,r_0)}^\pi v(s) - \sum_{a\in\mathcal{A}}\pi_s(a)(\alpha_{s,a}^r + \gamma\alpha_{s,a}^P\|v\|) \text{ for all } s \in \mathcal{S}.$$

This derivation enables us to derive an R$^2$ Bellman evaluation operator for the $(s,a)$-rectangular case. Indeed, the R$^2$ regularization function now becomes $\Omega_{v,\text{R}^2}(\pi_s) := \sum_{a\in\mathcal{A}}\pi_s(a)(\alpha_{s,a}^r + \gamma\alpha_{s,a}^P\|v\|)$, which yields the following R$^2$ operator:

$$[T^{\pi,\text{R}^2}v](s) := T_{(P_0,r_0)}^\pi v(s) - \Omega_{v,\text{R}^2}(\pi_s), \quad \forall s \in \mathcal{S}.$$

We aim to show that we can find a deterministic policy $\pi^d \in \Delta_{\mathcal{A}}^{\mathcal{S}}$ such that $[T^{\pi^d,\text{R}^2}v](s) = [T^{*,\text{R}^2}v](s)$ for all $s \in \mathcal{S}$. Given an arbitrary policy $\pi \in \Delta_{\mathcal{A}}^{\mathcal{S}}$, we first rewrite:

$$[T^{\pi,\text{R}^2}v](s) = r_0^\pi(s) + \gamma P_0^\pi v(s) - \Omega_{v,\text{R}^2}(\pi_s)$$

$$= \sum_{a\in\mathcal{A}}\pi_s(a)r_0(s,a) + \gamma\sum_{a\in\mathcal{A}}\pi_s(a)\langle P_0(\cdot|s,a),v\rangle - \left(\sum_{a\in\mathcal{A}}\pi_s(a)(\alpha_{s,a}^r + \gamma\alpha_{s,a}^P\|v\|)\right)$$

$$= \sum_{a\in\mathcal{A}}\pi_s(a)\left(r_0(s,a) - \alpha_{s,a}^r + \gamma(\langle P_0(\cdot|s,a),v\rangle - \alpha_{s,a}^P\|v\|)\right)$$

By [30, Lemma 4.3.1], we have that:

$$\sum_{a\in\mathcal{A}}\pi_s(a)\left(r_0(s,a) - \alpha_{s,a}^r + \gamma(\langle P_0(\cdot|s,a),v\rangle - \alpha_{s,a}^P\|v\|)\right)$$

$$\leq \max_{a\in\mathcal{A}}\left\{r_0(s,a) - \alpha_{s,a}^r + \gamma(\langle P_0(\cdot|s,a),v\rangle - \alpha_{s,a}^P\|v\|)\right\},$$

and since the action set is finite, there exists an action $a^* \in \mathcal{A}$ reaching the maximum. Setting $\pi^d(a^*) = 1$ thus gives the desired result. We just derived a regularized formulation of robust MDPs

with $(s, a)$-rectangular uncertainty set and ensured that the corresponding $R^2$ Bellman operators yield a deterministic optimal policy. In that case, the optimal $R^2$ Bellman operator becomes:

$$[T^{*,R^2}v](s) = \max_{a \in \mathcal{A}} \left\{ r_0(s,a) - \alpha^r_{s,a} + \gamma(\langle P_0(\cdot|s,a), v \rangle - \alpha^P_{s,a}\|v\|) \right\}.$$

$\square$

# D    Numerical experiments

Table 3: Hyperparameter set to obtain the results from Table 1

| Number of seeds per experiment | 5 |
|---|---|
| Discount factor $\gamma$ | 0.9 |
| Convergence Threshold $\theta$ | 1e-3 |
| Reward Radius $\alpha$ | 1e-3 |
| Transition Radius $\beta$ | 1e-5 |

In the following experiment, we play with the radius of the uncertainty set and analyze the distance of the robust/$R^2$ value function to the vanilla one obtained after convergence of MPI. Except for the radius parameters of Table 3, all other parameters remain unchanged. In both figures 1 and 2, we see that the distance norm converges to 0 as the size of the uncertainty set gets closer to 0: this sanity check ensures an increasing relationship between the level of robustness and the radius value. As shown in Fig. 1, the plots for robust MPI and $R^2$ MPI coincide in the reward-robust case, but they diverge from each other as the transition model gets more uncertain. This does not contradict our theoretical findings from Thms. 3.1-4.1. In fact, each iteration of robust MPI involves an optimization problem which is solved using a black-box solver and yields an approximate solution. As such, errors propagate across iterations and according to Fig. 2, they are more sensitive to transition than reward uncertainty. This is easy to understand: as opposed to the reward function, the transition kernel interacts with the value function at each Bellman update, so errors on the value function also affect those on the optimum and vice versa.

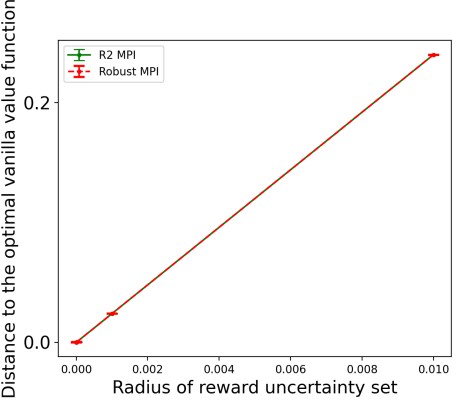

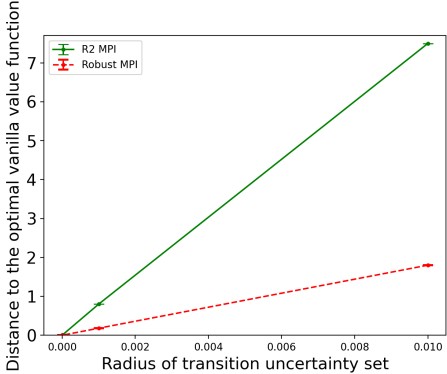

Figure 1: Distance norm between the optimal robust/$R^2$ value and the vanilla one as a function of $\alpha$ ($\beta = 0$) after 5 runs of robust/$R^2$ MPI

Figure 2: Distance norm between the optimal robust/$R^2$ value and the vanilla one as a function of $\beta$ ($\alpha = 0$) after 5 runs of robust/$R^2$ MPI