# OpenReview forum: "Twice regularized MDPs and the equivalence between robustness and regularization"
_NeurIPS.cc/2021/Conference — NeurIPS 2021 Poster_

### Official Review · Reviewer_sP9P · 2021-07-15

**Rating:** 5
**Confidence:** 3

**Summary:**

This paper formalized robust MDPs as general MDP with regularization. They first show that regularized MDPs are equivalent to MDPs with uncertain rewards. They also derive a regularized MDP formulation for robust MDPs with uncertain transitions. And they finally generalize a regularized MDP that can handle both reward and transition uncertainty. They argue that their regularized MDP framework provides better computational complexity and scalability compared to traditional robust optimization techniques.

**Limitations And Societal Impact:**

Limitations and impacts are addressed.

**Main Review:**

In lines 58-64, many quantities like v, \mathcal{U}, \mu etc. start appearing from no where. They were not defined before. This section needs reorganization making sure that everything is defined before referred or used.

It is possible to solve the reward robust MDP as a linear programming problem. The paper presents a policy gradient algorithm for the same purpose. How do they compare?

The theories presented in the paper look good from a high level. Though a rigorous verification of the correctness was not possible due to time constraint.

It is understandable that the paper is mostly theoretical, but one big weak point is not having any empirical analysis of the methods. How about having an empirical evaluation for robust and proposed regularized methods, possibly with different RL regularizers?


**Time Spent Reviewing:**

6

---

> ### Author Response · Authors · 2021-08-10
> **Response to Reviewer sP9P**
>
> Thank you for your review. As you note, our contribution is mostly theoretical, “interesting and significant”, to quote Reviewer FBfb. We respectfully ask you to reconsider your rating. Our work rigorously relates robustness to regularization, something that has only been hunched or empirically noticed before. This connection enables us to avoid solving recurrent min-max problems and reduces most robust MDPs to regularized MDPs.
>
>
> We address your comments in order.
>
> - The quantities used in l. 58-64 are introduced in the text. The notation $\mathcal{U}$ is defined as the uncertainty set in l. 57; $v$ is the variable to optimize as formulated in $\text{(RO)}$ and $\text{(CO)}$, while $\mu_0$ is a constant vector in $\mathbb{R}^{\mathcal{S}}$ denoting the initial state distribution. This last quantity is made explicit in l.104, but we shall briefly define it in the paragraph of interest.
>
> - “It is possible to solve the reward robust MDP as a linear programming problem”: A priori not. As we can see in Prop. 3.1, solving reward robust MDPs involves a robust optimization problem, itself deriving convex programming. Thm. 3.1 represents a contribution in that sense since it reformulates reward robust MDPs as linear programming (LP). In fact, the proof is based on a generalization of LP duality (see Appx. A.2), which is a key part for deriving this LP formulation. Our policy gradient theorem then follows from the linear constraints we obtained from Thm 3.1., just like the R2 Bellman operators we introduce next in the paper.
>
> - We ran MPI on a small 5x5 grid world, in order to evaluate the computing time of each approach. We used 3 baselines: MPI, robust MPI, and R2-MPI (see table below). For robust MPI, we used ball constrained uncertainty sets, in order to get a closer comparison with R2-MPI, whose limit point corresponds to the robust value function with a ball uncertainty set. The time spent for each algorithm until convergence were 0.006 seconds, 262 seconds, and 0.15 seconds, respectively. This corresponds to our expectations: R2-MPI is ~25 times slower than standard MPI since an additional regularization term needs to be computed at each iteration, but ~1746 times faster than robust MPI. Indeed an optimization problem needs to be solved at each Bellman update, which can take polynomial time for general convex programs. As an additional check-up, R2-MPI and robust MPI converge to the same policy and value function, which confirms the equivalence between the two approaches while highlighting the computational advantage of R2-MPI over robust MPI. We will add all of those results in a revised version of the paper.
>
> |                | MPI       | R2-MPI   | Robust MPI |
> |----------------|-----------|----------|------------|
> | Computing time | 0.006 sec | 0.15 sec | 262.3 sec  |

---

### Official Review · Reviewer_FBfb · 2021-07-16

**Rating:** 7
**Confidence:** 3

**Summary:**

The paper studies the connection between robust and regularized MDPs. The authors first show that solving robust MDPs with uncertain rewards is equivalent to solving regularized MDPs with a suitable policy-dependent regularization. Then, they extend this result to MDPs with uncertain transitions, obtaining the equivalence with both policy and value regularization. Finally, they propose R^2 MDPs, i.e., MDPs with both value and policy regularization for which, under certain assumptions, they guarantee good properties (eg contraction) for the corresponding Bellman operators and thus convergence of a modified policy iteration scheme. This allows solving robust MDP problems with the same computational cost as regularized ones.

**Limitations And Societal Impact:**

The authors discuss some of the limitations of their work, while there might be others that could be mentioned (see above). I do not see any societal impact.

**Main Review:**

Significance, novelty, and relevance
---

The paper considers a relevant problem, that of robust RL and, in particular, planning in robust MDPs, where reducing the computational complexity of Bellman updates is an important open area of research. While there have been many works that addressed this topic and the related one of regularized MDPs, to my knowledge, the contributions of this paper are novel. Overall I found this contribution interesting and significant.


Relation with prior work
---

To my knowledge, relevant prior works are discussed.


Clarity of writing
---

I found the paper well-written and easy to read. I think the authors did a good job in discussing and providing intuition for all results.


Detailed comments/questions
---

1. One limitation of this work is that no numerical result is provided. I know this is mostly theoretical work, but, since the final impact is on algorithms for solving robust MDPs (and a new algorithm is even provided), it would be good to see at least some experiments. For instance, it would be good to compare the MPI scheme for learning robust policies of Section 5 with existing robust DP methods (both in terms of convergence speed and computational cost), or to test the policy gradient derived in Section 3, possibly comparing with the existing related techniques mentioned in that section.

2. Theorem 5.1 proves that the optimal policy for R^2 MDPs is unique and, I assume, such a policy could be stochastic due to the regularization term. On the other hand, if I am not mistaken, for robust MDPs an optimal deterministic policy exists under certain conditions. So are there problems where there exists an optimal deterministic robust policy but solving the corresponding R^2 MDP we find only a stochastic one? If this is the case, it could be a potential limitation of using regularized methods, as in practice deterministic policies are desirable and we might not be able to find them using this alternative technique.

3. I appreciated that the authors showed (in Section 3) what are the uncertainty sets corresponding to common regularizers adopted in the literature. However, do the uncertainty sets presented in Section 3.2 make any sense in the robust RL literature? They only depend on the policy for which we are evaluating the regularizer and not on other MDP-related quantities (such as how uncertain we are about the rewards etc.). Some policy-dependent uncertainty sets were studied in [1], but still I do not see any connection with those reported here.

4. This paper considers finite states and actions, while most of the adopted "functional/operator-based" notation can be used also for more general state-action spaces. Do the results presented here easily extend to, say, countably infinite or continuous state-action spaces?

5. Line 132 mentions that regularized MDPs are standard MDPs with a modified reward function. Still, it should be mentioned that, while it is true that we can modify the expected reward r^\pi(s) of any policy \pi by including the regularizer, we cannot do the same for the original reward r(s,a) without making it policy-dependent (e.g., I am thinking about the standard entropy regularizer).

6. R^2 MDPs are introduced only for ball-constrained uncertainty sets, which yields the convenient norm-based regularization terms of Corollary 4.1. What about more general regularizers/uncertainty sets (e.g., KL-based)? Is it possible to obtain similar results?

7. The contraction property in Prop. 5.1 is wrt to 1-\epsilon_\star and not \gamma. So, while there is a computational gain wrt solving a robust MDP using classical techniques, could there be that the convergence rate of Bellman backups in the R^2 regularized MDP is much slower than in a standard regularized/unregularized MDP?


Minor:
- Line 99: is q a typo?
- Line 105: the notation for \Delta with both superscript and subscript was not introduced
- Prop. 3.2: is the subscript R^2 of J a typo? Also, in the main result there it is not clear under what MDP \mu_\pi is computed (though later the text mentions it is wrt the nominal model)

[1] Tirinzoni, Andrea, and Xiangli Chen. "Policy-conditioned uncertainty sets for robust Markov decision processes." Advances in neural information processing systems (2018).

**Time Spent Reviewing:**

4

---

> ### Author Response · Authors · 2021-08-10
> **Response to Reviewer FBfb**
>
> Thank you for your thorough review and your insightful questions.
>
> 1. We ran MPI on a small 5x5 gridworld, in order to compare the computing time and the convergence rate of each approach. We used 3 baselines: standard MPI, robust MPI and R2-MPI, which we ran on an Intel(R) Core(TM) i7-1068NG7 CPU @ 2.30GHz. The time taken for each algorithm until convergence were 0.006 seconds, 262 seconds and 0.15 seconds, respectively (see table below). This corresponds to our expectations: R2-MPI is slightly slower than standard MPI since an additional regularization term needs to be computed at each iteration; while robust MPI takes the longest. Indeed an optimization problem needs to be solved at each Bellman update, which can take polynomial time for general convex programs.
> |                | MPI       | R2-MPI   | Robust MPI |
> |----------------|-----------|----------|------------|
> | Computing time | 0.006 sec | 0.15 sec | 262.3 sec  |
> 2. You are correct. In robust MDPs, there exists an optimal deterministic policy as long as the uncertainty set is $(s,a)$-rectangular, i.e., independently defined for each state-action pair. When the uncertainty set is only $s$-rectangular, i.e., coupled across actions but still independently defined for each state, an optimal policy is generally stochastic [1, Table 1]. Our main results (Thms. 3.1 and 4.1) along with the R2 MDP setting build upon this more general $s$-rectangular case, which can indeed result in a stochastic optimal policy. Yet, when the uncertainty set is $(s,a)$-rectangular, we recover an optimal deterministic policy in the corresponding R2 MDP, just like robust MDPs. We shall add that remark to the paper.
> 3. In order to derive policy-independent uncertainty sets, we need to study the optimization problem that yields the optimal robust value function:
>
> $\max_{v\in\mathbb{R}^{\mathcal{S}}} \langle v, \mu_0 \rangle  \text{ s. t. } v\leq T_{(P,r)}^{\pi}v \text{ for all }   (P,r)\in\mathcal{U}\text{ and } \pi\in\Delta_{\mathcal{A}}^{\mathcal{S}},$
>
> then derive a similar result as Thm 3.1 to obtain policy-independent uncertainty sets for the reward. We deliberately describe our formulation for policy evaluation because we think this yields a clearer presentation and motivation for the R2 MDP setting. Yet, for completeness, we can add the policy-independent case in the Appendix and refer to it in the text body.
>
> 4. Our results easily extend to continuous but compact action spaces, as in standard MDPs [2]. Extension to countable/continuous state-space would be more involved because of the state-dependent regularization function. In fact, it would be interesting to study the R2 MDP setting under function approximations, and analyze how the regularizer can be derived under such approximation, but this is beyond the scope of the current paper.
> 5. Right. Good point. We will mention it in the paper.
> 6. Thms 3.1 and 4.1 apply for general uncertainty sets, while both statements further simplify for ball constrained uncertainty sets. This latter case corresponds to the R2 MDP formulation, which we believe can generalize to KL-based uncertainty sets: to do so, we need to compute the support function of KL-based sets. We should note that KL constraints only make sense for uncertain transition models, i.e., for probability distributions and not for reward functions. Although we just described the case of KL divergence, we could think of other uncertainty sets such as those based on negative entropy, Tsallis entropy, etc.
> 7. The quantity $\epsilon^*$ comes from Assumption 5.1. Therefore, as mentioned there and in l.280-294, there is an intrinsic dependence between $\gamma$ and $\epsilon^*$: when $\gamma \rightarrow 0$, standard Bellman updates yield faster convergence. In parallel, as $\gamma \rightarrow 0$, the value of $\epsilon$ required for Asm. 5.1 to hold increases, which makes the contracting coefficient $1-\epsilon$ tend to 0 as well, and the convergence rate of R2 Bellman updates is similar to the standard ones.
>
> Minor comments:
> - Yes, it should be $\mathbf{y}$. Thank you for noticing.
> - Although it can be deduced from the Notations paragraph, we shall introduce it explicitly there.
> - Yes, it should be $J_{\mathcal{U}}$. Also, we shall explicitly precise that $\mu_{\pi}$ is w.r.t. the nominal model inside the result statement.
>
> [1] Wiesemann, Wolfram, Daniel Kuhn, and Berç Rustem. "Robust Markov decision processes." Mathematics of Operations Research 38.1 (2013): 153-183.
>
> [2] Puterman, Martin L. Markov decision processes: discrete stochastic dynamic programming. John Wiley & Sons, 2014.

---

> > ### Comment · Reviewer_FBfb · 2021-09-01
> > **Re: Response to Reviewer FBfb**
> >
> > Dear authors,
> >
> > Thank you for your detailed response. It clarified all my doubts. Given also the new numerical simulations, I am increasing my score to 7.

---

### Official Review · Reviewer_NyMe · 2021-07-16

**Rating:** 7
**Confidence:** 2

**Summary:**

The paper highlights two limitations of current approaches for Robust RL:

1. Robust optimization methods are computationally demanding.
2. They do not account for uncertainty in the model dynamics.

The paper proposes to work around these limitations by learning robust MDPs using regularization. Their contributions are the following:

1. Showing that regularized MDPs are an instance of Robust MDPs with uncertain rewards; thus, policy iteration on reward-robust MDPs has the same time complexity as on regularized MDPs
2. Extend point 1 to MDPs with the uncertain transition.
3. Generalize regularized MDPs to twin regularized MDPs to retrieve robust MDPs.

**Limitations And Societal Impact:**

Limitations And Societal Impact are addressed.

**Main Review:**

Strengths:

1. The paper is well-contained. I am not familiar with Robust RL and even less familiar with work in theoretical RL. Still, I had no difficulty reading the paper and understanding its central claims/contributions.  Kudos to the authors.
2. I think the theoretical contributions are substantial and useful - connecting regularized MDPs to robust MDPs, extending the approach for uncertain dynamics function, etc. Though I should highlight that robustness/theory is not my area of expertise, so I might be over-estimating this contribution.

Questions / suggestions for improvement

1. Line 170 mentions a nominal model. How do we get access to this model? Is it part of the problem formulation, or is it learned using the sampled trajectories? If it is learned, how do we ensure that the nominal model is correct?

2. The paper mentioned that one of the limitations of existing approaches is that they are computationally demanding. I may have missed it, but it seems that the paper did not include any evidence for the twin regularized MDPs to be computationally more efficient.  Line 177 mentions that "policy iteration on reward-robust MDPs has the same convergence rate as regularized MDPs, which in turn is the same as standard MDPs." Still, it would be helpful to include both the convergence rates and error bounds in the main paper (for both the proposed approach and the standard robustness-based approaches) to provide a complete comparison. For example, regularized MDPs are a particular case of robust MDPs. So other robustness methods may have a slower convergence rate but achieve better performance. The paper can be improved by including these tradeoffs explicitly.

3. Missing empirical evidence - I understand that this paper is focusing on theoretical contributions. It may strengthen the paper to include some experimental results as well (even if it is on simplistic grid worlds).

**Time Spent Reviewing:**

4

---

> ### Author Response · Authors · 2021-08-10
> **Response to Reviewer NyMe**
>
> Thank you for considering the quality of our writing and for the positive assessment of our contributions. We address each of your questions in order.
>
> 1. The nominal model can typically be an empirical (e.g., maximum likelihood) estimate derived from sampled trajectories. As such, nothing ensures its correctness, i.e., it can be different from the simulator’s parameters. In fact, a provably optimal policy for this nominal can be very sensitive to small variations and/or errors on the MDP model [1]. This constitutes the main motivation for taking a robust approach and accounting for confidence intervals around that model estimate. Alternatively, the nominal model can be part of the problem formulation: the simulator’s transition and reward functions are generally unknown, which may necessitate robust learning.
>
> 2. Consider the value iteration (VI) algorithm, a particular case of MPI: both standard VI and robust VI have a geometric rate of convergence, just like VI for R2 MDPs. However, in robust VI, each iteration involves a max-min problem, which requires polynomial time computation for solving a convex program. Differently, VI in R2 MDPs only involves a maximization problem to find a greedy policy, which can be solved within linear time [2]. We will add this tradeoff to the text.
>
> 3. We followed your suggestion and ran MPI on a 5x5 gridworld domain. Our goal was to compare the computation time required for MPI to converge. We used three baselines: standard (non-robust) MPI, R2 MPI and Robust MPI, which we ran on an Intel(R) Core(TM) i7-1068NG7 CPU @ 2.30GHz. As expected, standard MPI takes the least time to converge (base = 0.006 seconds); R2 MPI takes slightly longer because of the additional computation of regularizers (25 time bases); robust MPI takes much longer because of the convex program involved at each iteration (43 666 time bases). As such, although R2-MPI is about 25 times slower than standard MPI, it speeds up robust MPI by a factor of 1746 approximately. We will add those results to the paper, which we display in the table below. We hope this answers your questions. Please feel free to ask us if it raises other ones.
>
> |                | MPI       | R2-MPI   | Robust MPI |
> |----------------|-----------|----------|------------|
> | Computing time | 0.006 sec | 0.15 sec | 262.3 sec  |
>
>
> [1] Mannor, Shie, et al. "Bias and variance approximation in value function estimates." Management Science 53.2 (2007): 308-322.
>
> [2] Duchi, John, et al. "Efficient projections onto the l 1-ball for learning in high dimensions." Proceedings of the 25th international conference on Machine learning. 2008.

---

> > ### Comment · Reviewer_NyMe · 2021-08-18
> > **Thank you for the response!**
> >
> > Could the authors provide a description for the grid world task (or link to the paper from where they are using the task, if it is a standard task).

---

> > > ### Author Response · Authors · 2021-08-19
> > > **Description of the Grid World task**
> > >
> > > The gridworld we ran experiments on is inspired from [1, Example 4.1], where the agent must reach either one of two goal states: the first one (G-left) is on the top left corner, and the second one (G-right) on the bottom right corner of the grid. The agent deterministically moves according to the chosen direction. In our version, the state-space is of size 5x5=25, while the reward value is 0 in non-goal states, 1 in G-left and 10 in G-right. In the robust version, the uncertainty sets for both reward and transition models are given by L2-balls constraints: the reward uncertainty set is a ball of radius of 0.02 around the true underlying reward function while the transition uncertainty is a ball of radius 0.01 around the true kernel. These norm constraints yield a second-order cone programming, which needs to be solved at each Bellman update.
> > >
> > > Please let us know if you need additional information.
> > >
> > > [1] Sutton, Richard S., and Andrew G. Barto. Reinforcement learning: An introduction. MIT press, 2018.

---

> > > > ### Comment · Reviewer_NyMe · 2021-08-31
> > > > **Thank you for the response!**
> > > >
> > > > Thank you for the additional experiment. I am increasing the score to 7.

---

### Official Review · Reviewer_qvyT · 2021-07-23

**Rating:** 8
**Confidence:** 2

**Summary:**

The paper establishes an equivalence between regularized MDPs and robust MDPs. The first main result shows that policy-regularized MDPs are equivalent to reward-robust MDPs, gives examples of equivalences between common regularizers and uncertainty sets, and gives a policy gradient algorithm for reward-robust MDPs. The second main result extends the equivalence to robust MDPs with both reward and transition uncertainty. The third main result introduces R^2 MDPs, which extend regularized MDPs with a value regularizer. It is shown that under certain assumptions on the uncertainty sets, the equivalent R^2 MDP can be solved with MPI.

**Limitations And Societal Impact:**

The paper clearly states the assumptions needed for its analysis, which seem to be the only significant limitations. I see no potential for negative societal impact.

**Main Review:**

## Originality

I am not familiar with the literature in this area, but the paper claims all 3 main results as original. Related work is covered thoroughly.

## Quality

I don't have the mathematical background to verify the analysis. The statements of the results are precise and seem to include all required assumptions. The work addresses both the narrower case of reward-robust MDPs as well as the case of general robust MDPs, and provides both a construction of the equivalent regularized problems and a solution algorithm.

## Clarity

The paper is clearly-written but is fairly dense. Mathematical notation is precise.

## Significance

Each of the main results seems to be significant on its own. The connection between regularization and robustness seems like it will be fertile for further analyses. The MPI algorithm for R^2 MDPs is significant in that it establishes that R^2 MDPs with appropriate assumptions have the same complexity as ordinary MDPs, making them a tractable subset of robust MDPs.

**Time Spent Reviewing:**

2

---

> ### Author Response · Authors · 2021-08-10
> **Response to Reviewer qvyT**
>
> Thank you very much for your appreciation of our writing and our contributions!

---

### Decision · Program_Chairs · 2021-09-27

**Decision:**

Accept (Poster)

**Comment:**

All but one reviewer recommend publication.  The one who does not is concerned about the lack of empirical results, but I think this is a topic of interest to the community and its reasonable to have a purely theoretical paper on this.